# Interchain supramolecular interactions drive nearly 21% efficiency organic solar cells

Wei Gao[1,7], Yulong Hai[2,7], Jinyan Zeng[1], Hao Xia[1] ✉, Ruijie Ma [3,4] ✉,
Top Archie Dela Peña [2], Jiaying Wu [2], Chunhui Duan [5], Jian-Xin Tang [6],
Zhanhua Wei [1] ✉ & Gang Li [3] ✉

A small-molecule acceptor, S-Cb, substituted with a cyclobutyl group that introduces high ring strain, was designed and synthesized. Thanks to the rigid and planar structure of cyclobutyl, S-Cb can form interchain supramolecular interactions through hydrogen bonding with L8-BO at the external side chains. This clamping effect not only effectively suppresses the electron-phonon coupling but also promotes the formation of high-quality acceptor alloy phases in the ternary active layer, thereby optimizing carrier behaviors and reducing non-radiative energy loss. The clamping effect reaches its maximum when S-Cb and L8-BO are in equal proportion, where organic solar cells (OSCs) based on D18:S-Cb:L8-BO achieved an impressive efficiency of 20.93%, with a certified efficiency of 20.74%. In summary, the cyclobutyl-mediated interchain supramolecular interactions suppress the electron-phonon coupling and optimize the acceptor alloy phase for efficient ternary OSCs.

Organic solar cells (OSCs) offer significant potential for applications in wearable electronics and building-integrated photovoltaics, which stems from their lightweight design, mechanical flexibility, semi-transparency, and compatibility with solution-based, large-area printing techniques[1–9]. However, compared to the most advanced silicon-based or organic-inorganic hybrid perovskite solar cells[10–12], the power conversion efficiency (PCE) of OSCs still lags significantly behind, which is mainly due to the strong electron-phonon coupling in organic semiconductor materials[13–20]. The electron-phonon coupling refers to the interaction between charge carriers and lattice vibrations. Charge carriers undergo scattering by phonons and exchange energy with the lattice, resulting in a rapid relaxation process that leads to undesirable non-radiative recombination[16,17]. Therefore, mitigating the electron-phonon coupling in the active layer is considered to be crucial for enhancing the photovoltaic performance of OSCs[14,21–24].

The electron-phonon coupling in organic semiconductors primarily originates from two distinct mechanisms: (i) the high-frequency intramolecular vibrations, such as C-C stretching within the conjugated backbone, and (ii) the low-frequency intermolecular motions, including out-of-plane bending and in-plane rotation of entire molecules. The coupling strength associated with specific intramolecular modes (i) can be directly quantified by the corresponding Huang-Rhys factor (S-factor). In contrast, the disordered and collective nature of the intermolecular motions (ii) makes their direct microscopic quantification challenging. Therefore, macroscopic structural parameters like the free volume ratio (FVR) of the active layer are often employed as effective proxies to gauge the extent of such molecular-scale

[1]Xiamen Key Laboratory of Optoelectronic Materials and Advanced Manufacturing, Institute of Luminescent Materials and Information Displays, College of Materials Science and Engineering, Huaqiao University, Xiamen, China. [2]Thrust of Advanced Materials, The Hong Kong University of Science and Technology (Guangzhou), Nansha, Guangzhou, China. [3]Department of Electrical and Electronic Engineering, Research Institute for Smart Energy (RISE), Photonic Research Institute (PRI), The Hong Kong Polytechnic University, Hong Kong, China. [4]Hangzhou International Innovation Institute, Beihang University, Hangzhou, China. [5]Institute of Polymer Optoelectronic Materials and Devices, Guangdong Basic Research Center of Excellence for Energy & Information Polymer Materials, State Key Laboratory of Luminescent Materials and Devices, South China University of Technology, Guangzhou, China. [6]Macao Institute of Materials Science and Engineering (MIMSE), Faculty of Innovation Engineering, Macau University of Science and Technology, Taipa, Macao, China. [7]These authors contributed equally: Wei Gao, Yulong Hai. ✉e-mail: xiahao919@hqu.edu.cn; ruijiema@buaa.edu.cn; weizhanhua@hqu.edu.cn; gang.w.li@polyu.edu.hk

thermal fluctuations[21]. Generally, a smaller FVR indicates more restricted molecular motion, which helps reduce electron-phonon coupling. Current small-molecule acceptors (SMAs) consist of a conjugated backbone with soluble alkyl chains around[25–44]. It is acknowledged that C-C bonds between sp²-hybridized C atoms are more robust due to higher bond energy, and alternated C-C single and double bonds in conjugated backbone form a delocalized π-structure, therefore, the rigidity of the conjugated backbone is significantly greater than that of alkyl chains linked by sp³-hybridized C atoms. As a result, C-C stretching vibrations in the large π-conjugated backbone of SMAs are much weaker than those in flexible alkyl chains. Additionally, the conjugated backbones of SMAs readily form π-π stacking in the solid state, which is able to effectively reduce thermal motion within the molecular backbone through π-π interactions. In view of this, we hypothesize that increasing the rigidity of alkyl chains in SMAs can effectively suppress electron-phonon coupling in the active layers of OSCs.

Compared to straight or branched alkyl chains, cyclic ones exhibit greater rigidity due to their ring strain[45]. As the ring size decreases, the angles between C-C single bonds become increasingly compressed to enhance ring strain and force cyclic alkyl chains to adopt a more planar conformation, which improves the rigidity of the alkyl chain to a certain extent. Among common cyclic alkyl chains, such as cyclohexyl, cyclopentyl, cyclobutyl, and cyclopropyl, the angles of the C-C bond are approximately 109°, 108°, 90°, and 60°, respectively. Cyclohexyl and cyclopentyl groups can largely maintain the ideal tetrahedral angle of 109.5° for sp³-hybridized carbon atoms, resulting in low ring strain. These rings adopt characteristic conformations—the non-planar envelope-like conformation for cyclopentane and the stable chair conformation for cyclohexane. In contrast, cyclopropyl exhibits the highest ring strain and a rigid planar geometry. Although its pronounced ring strain could, in principle, enhance structural rigidity, the cyclopropyl ring is prone to ring-opening reactions under various chemical conditions, including those encountered during the synthesis of Y6-type derivatives, which limits its practical utility in such materials. Cyclobutyl, with a C-C bond angle of about 90° that significantly deviates from the ideal tetrahedral angle, possesses considerable ring strain that imparts notable conformational rigidity. To partially relieve this strain, the cyclobutane ring adopts a slightly folded butterfly-like conformation, with a dihedral angle between the ring planes of only about 14°, indicating a relatively flat structure. This combination of substantial rigidity, moderate steric bulk, and sufficient synthetic stability makes the cyclobutyl group an attractive structural motif in the design of organic photovoltaic materials. Moreover, the structural characteristics of cyclobutane place it between flexible alkyl chains and rigid aromatic groups, ensuring molecular solubility while imparting moderate rigidity to the alkyl chain region.

Taking these into consideration, introducing a cyclobutyl group into the central core of SMAs is expected to reduce electron-phonon coupling in the active layer, primarily because, on one hand, the relatively rigid cyclobutyl can partially suppress C-C stretching vibrations of alkyl chains. On the other hand, cyclobutyl substitution may alter molecular packing modes of SMAs, potentially reducing the thermal motion of SMAs. Motivated by these insights, a new SMA, namely S-Cb with cyclobutyl groups substituted at the thiophene β-position was designed and synthesized. Cyclobutyl cyclization strategy can effectively increase the rigidity of S-Cb and reduce the Stokes shift in contrast to that of L8-BO, revealing a suppressed relaxation process caused by electron-phonon coupling. Simultaneously, the bandgap of S-Cb was narrowed along with molar extinction coefficients and crystallinity enhanced. It is discovered that the rigid cyclobutyl group on S-Cb can clamp with the bifurcated alkyl chains of L8-BO via hydrogen bonding. These interchain supramolecular interactions facilitate the formation of a highly ordered acceptor alloy phase, which significantly

reduces the S-factor and FVR of the ternary active layer and suppresses electron-phonon coupling for lower non-radiative energy loss ($E_{loss}$). Binary device based on D18:S-Cb achieved a PCE of 19.63%, slightly lower than that of D18:L8-BO-based device (19.85%). Ternary device based on D18:S-Cb:L8-BO gained an impressive PCE of 20.93% with a certified efficiency of 20.74% when the weight ratio of two acceptors was 1:1. This work demonstrates that the introduction of rigid cyclobutyl side chains facilitates the formation of interchain supramolecular interactions, thereby suppressing electron-phonon coupling and optimizing acceptor alloy phase for enhanced photovoltaic performance of ternary devices.

## Results

### Molecular structure and rigidity

Molecular structures of polymer donor D18 and investigated SMAs S-Cb and L8-BO are presented in Fig. 1a. The detailed synthetic routes for S-Cb is illustrated in Supplementary Fig. 1 and follow a procedure similar to that used for Y6 and its derivatives[41]. The intermediates and final product in the S-Cb synthesis were characterized using ¹H NMR, ¹³C NMR, and mass spectrometry (MS), confirming the correct chemical structures. Evaluation reveals that if L8-BO is synthesized via the same route as S-Cb, the alkyl chain combinations in both materials would lead to comparable large-scale production costs for S-Cb relative to L8-BO. It is worth noting that the (bromomethyl)cyclobutene, the precursor of carboxylic acid containing one extra carbon atom, required for S-Cb is commercially available at a lower price than the 2-butyloctyl bromide used for L8-BO. Thermal stability of S-Cb and L8-BO were assessed using thermogravimetric analysis (TGA) measurement, which revealed that 5% decomposition temperature ($T_d$) of S-Cb is 324 °C, approximately 10 °C lower than that of L8-BO (335 °C, Supplementary Fig. 2). This decrease in thermal stability is likely attributed to the inherent ring strain in cyclobutane unit, which facilitates ring opening under elevated thermal stress.

In order to evaluate the change in rigidity of SMAs upon cyclobutyl substitution, temperature-dependent UV-vis absorption spectra of S-Cb and L8-BO neat films were measured (Supplementary Fig. 3). The redshifts in the spectra after thermal annealing of as-cast films were analyzed to assess the thermal transition that can be quantified by the deviation metric ($DM_T$), helping determine the glass transition temperature ($T_g$) of SMAs[46]. $T_g$, defined as the lowest temperature at which a molecular segment starts to move, is positively correlated with molecular rigidity. Generally, the higher the rigidity, the higher the $T_g$. As shown in Fig. 1b, the $T_g$ values of L8-BO and S-Cb determined by the intersection of two linear fitting curves are found to be 87.2 °C and 92.8 °C, respectively. Although alkyl chains attached to the N-position of S-Cb's central core is longer than those of L8-BO, the $T_g$ of S-Cb can still be enhanced through side chain cyclization in the thiophene β-position, indicating that the cyclization strategy can effectively increase the rigidity of SMAs.

To gain insights into the effects of increased rigidity in SMAs on electron-phonon coupling, photoluminescence (PL) spectroscopy of pure L8-BO and S-Cb films were further measured (Fig. 1c). The PL peaks of L8-BO and S-Cb are 902.5 nm and 903.2 nm, respectively. By combining these values with absorption peaks (Supplementary Table 1), the Stokes shifts of L8-BO and S-Cb neat films were calculated to be 1420 cm⁻¹ and 1382 cm⁻¹, respectively. The reduction in the Stokes shift indicates a decrease in non-radiative $E_{loss}$ caused by relaxation process of excitons as they transition from the excited state to the ground state, suggesting electron-phonon scattering and energy exchange with the lattice are suppressed. Furthermore, the Full Width at Half Maximum (FWHM) of PL spectrum of S-Cb is 101.6 nm, significantly lower than that of L8-BO (111.9 nm). The narrowing of FWHM in S-Cb PL spectrum is attributed to a reduction in vibrational modes, which results from increased atomic ordering within S-Cb molecule,

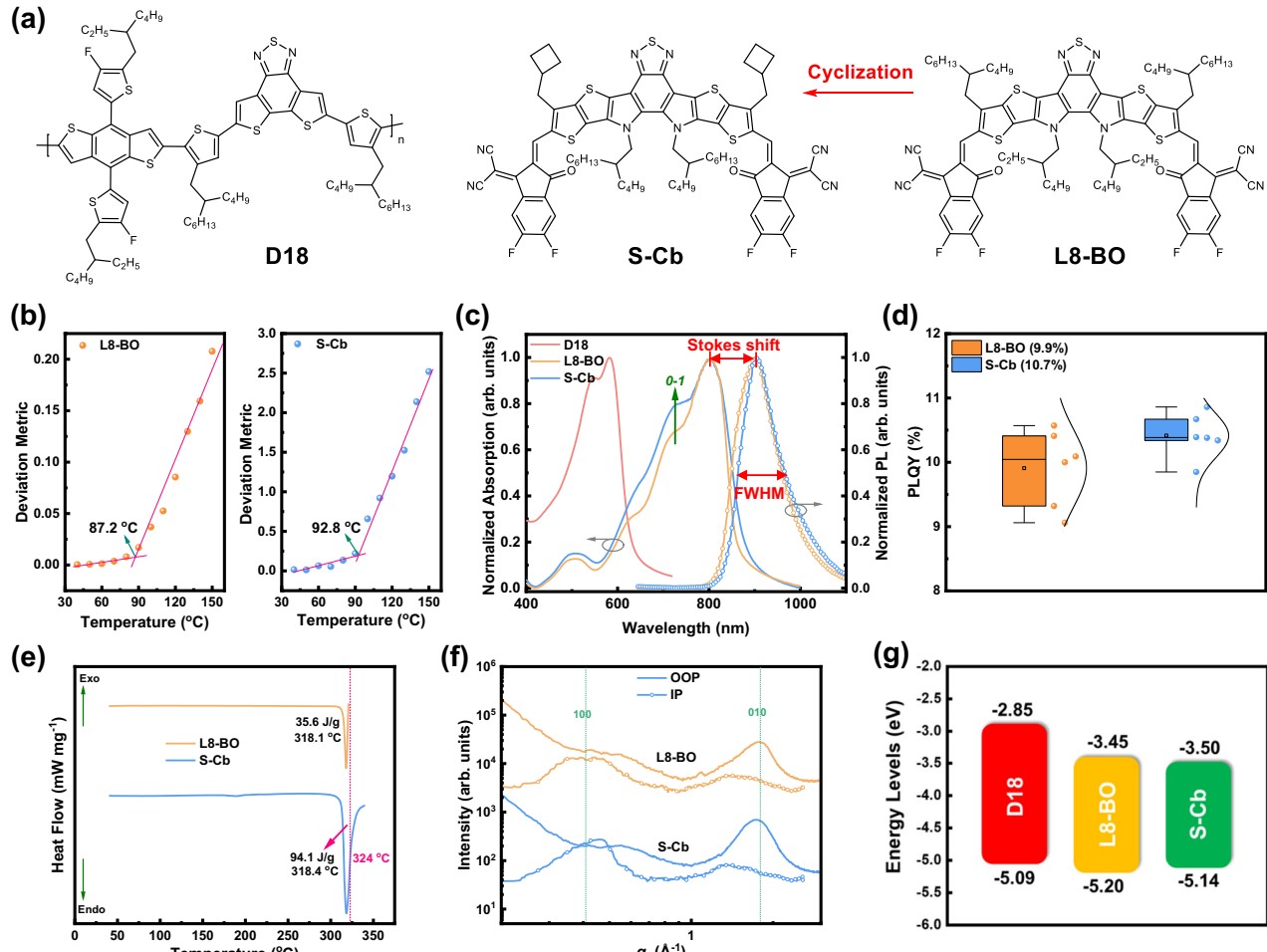

**Fig. 1 | Chemical structures and photophysical properties. a** Chemical structures of D18, S-Cb, and L8-BO. **b** Deviation metric from temperature-dependent UV-vis absorption spectra of S-Cb and L8-BO neat films. **c** Normalized absorption and PL spectra of S-Cb and L8-BO neat films. **d** PLQY of S-Cb and L8-BO neat films (the box plot elements are defined as follows: the center line represents the median; box limits represent the first and third quartiles; whiskers represent the maximum and minimum values within 1.5 times the interquartile range; the small square represents the mean value; the error bar represents the standard error of the mean with $n = 6$; individual data points show each measurement). **e** DSC curves of S-Cb and L8-BO (the red line is the $T_d$ of S-Cb). **f** 1D line-cut profiles from GIWAXS measurements of S-Cb and L8-BO neat films. **g** HOMO and LUMO energy levels of D18, S-Cb, and L8-BO films.

leading to fewer vibrational energy levels in the molecule[14]. PL experiments reveal that increasing the rigidity of alkyl chain in SMAs can effectively suppress electron-phonon coupling.

Parallel measurements of the photoluminescence quantum yield (PLQY) for L8-BO and S-Cb neat films were conducted, with the results from six independent measurements shown in Fig. 1d. The average PLQY was 9.9% for L8-BO and 10.7% for S-Cb. The slightly higher PLQY of S-Cb suggests that the cyclobutyl substituent helps to suppress electron-phonon coupling to reduce the non-radiative transition rate ($k_{nr}$) to some extent, according to the relationship PLQY = $\frac{k_r}{k_r + k_{nr}}$, where $k_r$ is the radiative transition rate.

To decouple whether this suppression originates from the inherent molecular rigidity or the molecular packing effects, we further examined their properties in dilute solution (Supplementary Fig. 4). At the same concentration ($3.0 \times 10^{-6}$ mmol mL$^{-1}$), the PL intensity of S-Cb is 2.4 times that of L8-BO. Comparison of normalized absorption and PL spectra reveals that S-Cb has a smaller Stokes shift (838 cm$^{-1}$ for S-Cb vs. 862 cm$^{-1}$ for L8-BO) and a narrower emission FWHM (55.9 nm for S-Cb vs. 61.0 nm for L8-BO). Most notably, the PLQY of S-Cb in dilute solution reaches 29.2%, nearly double that of L8-BO (16.6%). Considering the negligible molecular aggregation in dilute solution, these results strongly suggest that the inherent rigidity of the cyclobutyl group can effectively suppress intramolecular electron-

phonon coupling, thereby reducing exciton-molecular vibration interaction and mitigating $k_{nr}$ caused by vibrational relaxation.

We posit that this intramolecular vibrational suppression effect of the cyclobutyl group persists in the solid state. However, the transition from solution to film leads to a much more pronounced decrease in PLQY for S-Cb than for L8-BO, revealing that S-Cb undergoes a more significant aggregation-caused quenching (ACQ) effect in the solid state, likely attributable to the cyclobutyl substitution promoting a stronger propensity for *H*-aggregation as corroborated by the more distinct *O-1* vibrational shoulder in its absorption spectrum (Fig. 1c), which will exacerbate electron-phonon coupling and facilitate non-radiative energy dissipation[47]. The weaker overall electron-phonon coupling observed in S-Cb compared to L8-BO is likely primarily due to the effective reduction of molecular vibrations by the rigid cyclobutyl group. Simultaneously, its shorter alkyl chain might also enhance molecular packing (particularly *H*-aggregation), which exacerbates ACQ and partially counteracts the benefits gained from rigidity.

## Photophysical and crystallization properties
Subsequently, impacts of cyclobutyl cyclization on molecular absorption, crystallinity, and energy levels were investigated. The normalized UV-Vis absorption spectra of S-Cb and L8-BO in dilute solution and neat films are shown in Supplementary Fig. 5 and Fig. 1c,

respectively. Compared to L8-BO, S-Cb exhibits a slight red shift of approximately 2 nm in solution. To quantify their extinction coefficients, five different concentrations of dilute S-Cb and L8-BO solutions were prepared, and their absorption spectra were measured (Supplementary Fig. 6). The plots of maximum absorbance versus concentration yielded molar extinction coefficients of $2.15 \times 10^5$ and $2.04 \times 10^5$ $M^{-1}$ $cm^{-1}$ for S-Cb and L8-BO, respectively. In the solid state, S-Cb shows a more pronounced *O-1* vibronic shoulder and a narrower optical bandgap (1.36 eV for S-Cb vs. 1.38 eV for L8-BO), indicating that the cyclobutyl substitution enhances the molecular packing behavior of SMA. Additionally, the absorption spectra of S-Cb and L8-BO films with varying thicknesses were measured (Supplementary Fig. 7). The fitted absorption coefficients for S-Cb and L8-BO films were $4.49 \times 10^4$ and $3.97 \times 10^4$ $cm^{-1}$, respectively (Supplementary Table 1). Overall, cyclobutane-substitution improves the absorption properties of S-Cb.

Differential scanning calorimetry (DSC) measurements were performed for S-Cb and L8-BO to investigate the crystallization behavior of two acceptor molecules. As shown in Fig. 1e, both SMAs exhibit distinct melting peaks near their decomposition temperatures, with melting temperatures ($T_m$) of 318.4 °C for S-Cb and 318.1 °C for L8-BO. However, the enthalpy change ($\Delta H_m$) associated with the melting peak of S-Cb (94.1 J/g, determined by integrating the area of the melting peak only up to the red line, and the portion after the red line was excluded because the thermal effect in that interval includes additional heat absorption due to the thermal decomposition of S-Cb) is substantially higher than that of L8-BO (35.6 J/g), indicating that cyclobutyl substitution markedly enhances the intermolecular interactions and thus crystallization tendency of S-Cb.

Grazing-incidence wide-angle X-ray scattering (GIWAXS) measurements were conducted on neat films of S-Cb and L8-BO to gain further insights into how the cyclobutyl substitution affect molecular packing behavior of S-Cb. The corresponding 2D GIWAXS patterns and 1D line-cut profiles are shown in Supplementary Fig. 8 and Fig. 1f, respectively. Both S-Cb and L8-BO films exhibit well-defined face-on orientations. Notably, the (010) and (100) diffraction peak intensities of S-Cb are stronger than those of L8-BO, suggesting enhanced crystallinity in the S-Cb film. Analysis of the 1D profiles reveals that the π-π stacking distance of S-Cb (3.62 Å) is slightly larger than that of L8-BO (3.57 Å), likely due to the larger steric hindrance introduced by the 2-butyloctyl group attached to the *N* atom of S-Cb. However, S-Cb shows a longer π-π coherence length (CL of 12.05 Å for S-Cb and 10.92 Å for L8-BO), indicating a more ordered π-π stacking arrangement. Furthermore, the lamellar packing distance and CLs for S-Cb are 13.78 Å and 32.40 Å, respectively-both significantly improved compared to those of L8-BO (14.61 Å and 22.58 Å). These results suggest that cyclobutyl substitution primarily enhances crystallinity by promoting lamellar packing, instead of intermolecular π-π stacking. Electron mobilities of neat S-Cb and L8-BO films were measured by using the space-charge limited current (SCLC) method. By fitting the current density-voltage (*J-V*) curves (Supplementary Fig. 9), the electron mobilities were determined to be $2.44 \times 10^{-4}$ and $2.88 \times 10^{-4}$ $cm^2$ $V^{-1}$ $s^{-1}$ for S-Cb and L8-BO, respectively. The slightly lower electron mobility of S-Cb is consistent with the GIWAXS results, as the steric hindrance from the 2-butyloctyl group in S-Cb increases the π-π stacking distance, thereby impeding charge transport.

The highest occupied molecular orbital (HOMO) energy levels of D18, L8-BO, and S-Cb neat films were determined by ultraviolet photoelectron spectroscopy (UPS), as shown in Supplementary Fig. 10. The measured HOMO levels ($HOMO^{UPS}$) are −5.09 eV, −5.20 eV, and −5.14 eV for D18, L8-BO, and S-Cb, respectively. To accurately calculate the corresponding lowest unoccupied molecular orbital (LUMO) levels, cyclic voltammetry (CV) measurements were further performed on L8-BO and S-Cb neat films (Supplementary Fig. 11). The HOMO/LUMO levels derived from CV curves ($HOMO^{CV}/LUMO^{CV}$) are −5.63/−3.88 eV for L8-BO and −5.57/−3.93 eV for S-Cb. The fundamental gap

($E_g^{fund} = E_{LUMO}^{CV} − E_{HOMO}^{CV}$) is thus determined to be 1.75 eV for L8-BO and 1.64 eV for S-Cb, while that of D18 (2.24 eV) is taken from our previous report[40]. Consequently, the LUMO levels were calculated using $E_{LUMO} = E_{HOMO}^{UPS} + E_g^{fund}$, yielding values of −2.85 eV for D18, −3.45 eV for L8-BO, and −3.50 eV for S-Cb. It is noteworthy that the $E_g^{fund}$ physically represents the minimum energy required to completely remove an electron from the HOMO level, corresponding to the difference between the ionization energy (IE) and the electron affinity (EA). Therefore, using $E_g^{fund}$ instead of the optical gap ($E_g^{opt}$) provides a more rigorous basis for estimating the LUMO level. The trend in HOMO and LUMO levels revealed by UPS and CV is consistent, providing mutual verification. Moreover, the energy levels of L8-BO and S-Cb align well with those of D18 (Fig. 1g), which is expected to create an effective driving force for charge separation at the donor/acceptor interface. However, the slightly lower LUMO level of S-Cb compared to that of L8-BO may adversely affect the enhancement of the $V_{OC}$.

## Photovoltaic performance

The photovoltaic performance of S-Cb was evaluated by fabricating OSCs devices with an architecture of indium tin oxide (ITO)/poly(3,4-ethylenedioxythiophene):poly(styrenesulfonate) (PEDOT:PSS)/active layer/poly(9,9-bis(3-(*N,N*-dimethyl)-*N*-ethylammonium-propyl)−2,7-fluorene-*alt*−2,7-(9,9-dioctylfluorene)) dibromide:melamine (PFN-Br:MA)[48]/Ag. Detailed fabrication procedures are provided in the Methods section, and *J-V* curves of optimized device are shown in Fig. 2a with corresponding photovoltaic parameters are summarized in Table 1. The D18:S-Cb-based OSCs achieved a $V_{OC}$ of 0.888 V, a $J_{SC}$ of 27.57 mA $cm^{-2}$, and an FF of 80.2%, yielding a PCE of 19.63%. For comparison, the D18:L8-BO-based device showed a slightly higher PCE of 19.85%, with a $V_{OC}$ of 0.925 V, a $J_{SC}$ of 26.83 mA $cm^{-2}$, and an FF of 80.0%. In view of complementary photovoltaic parameters of D18:S-Cb and D18:L8-BO devices, a ternary blending strategy was considered with L8-BO introduced as a third component into D18:S-Cb blend to form ternary devices. As a result, a significant performance enhancement was observed. Upon adding 25% weight ratio of L8-BO, the D18:S-Cb:L8-BO device was able to achieve a PCE of 20.44%, a great improvement in photovoltaic performance compared to that of D18:S-Cb-based devices, which is attributed to simultaneous improvements in $V_{OC}$, $J_{SC}$, and FF. When L8-BO content was increased to 50%, the PCE of D18:S-Cb:L8-BO-based devices can be further improved, resulting in an impressive PCE of 20.93%, along with a third-party certified efficiency as high as 20.74% (a $V_{OC}$ of 0.913 V, a $J_{SC}$ of 27.90 mA $cm^{-2}$, and an FF of 81.4%, Fig. 2b and Supplementary Fig. 12). When L8-BO ratio was increased to 75%, although $V_{OC}$ continued to rise, both $J_{SC}$ and FF declined, leading to a reduced PCE of 20.40%. 10 to 15 data points were collected for these OSC devices, as shown in Supplementary Fig. 13 (Based on sample size, mean, and standard deviation, F- and t-test statistical analysis of the photovoltaic parameters across different L8-BO content indicated no statistically significant differences between the groups). The average PCEs and their standard deviation are presented in Table 1, demonstrating good device reproducibility.

To further verify the general applicability of S-Cb as a ternary component, we replaced L8-BO with another high-performance acceptor, BTP-eC9, to fabricate ternary devices based on D18:S-Cb:BTP-eC9 with a BTP-eC9 content of 50 wt%. The *J-V* curves and EQE spectra of the champion device are presented in Supplementary Fig. 14, with the corresponding photovoltaic parameters summarized in Supplementary Table 2. The binary device based on D18:BTP-eC9 yielded a PCE of 19.21%, slightly lower than that of D18:S-Cb-based OSCs. In contrast, the ternary device based on D18:S-Cb:BTP-eC9 (1:1) achieved an enhanced PCE of 20.23%, with a $V_{OC}$ of 0.876 V, a $J_{SC}$ of 28.54 mA $cm^{-2}$, and an FF of 80.9%. The efficiency enhancement, primarily attributed to the notable improvement in FF, represents an absolute increase of approximately 1%. The magnitude of this gain is comparable to that observed when introducing S-Cb into the D18:L8-

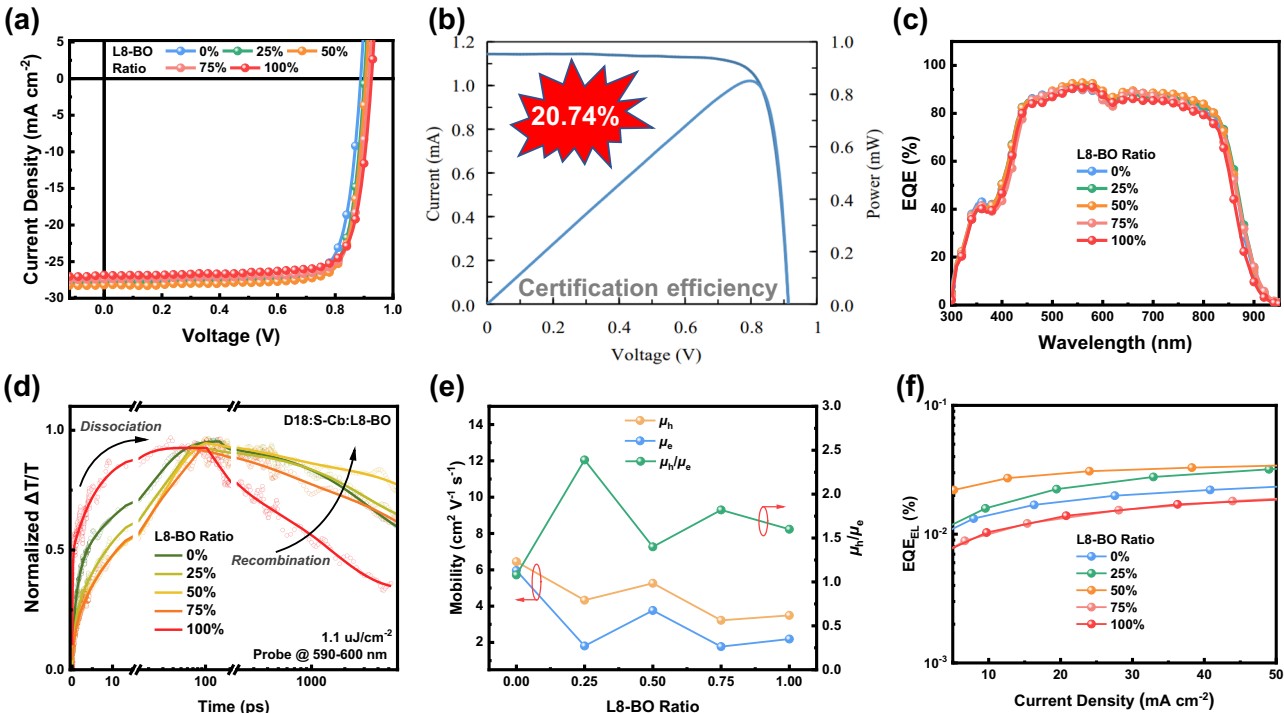

**Fig. 2 | Photovoltaic performance. a** *J*-*V* curves of the optimized devices with different L8-BO contents. **b** Certified efficiency. **c** EQE spectra of the corresponding devices. **d** Free charge generation and recombination kinetics of D18:S-Cb:L8-BO active layers with varying L8-BO ratios. **e** Electron and hole mobilities of active layers with different L8-BO contents. **f** EQE$_{EL}$ curves of active layers with varying L8-BO ratios.

BO binary system. These results demonstrate that cyclobutyl-substituted S-Cb serves as a versatile ternary component, exhibiting excellent compatibility with acceptor materials bearing either linear or branched alkyl side chains and effectively boosting device performance.

The EQE spectra of optimal OSCs based on D18:S-Cb:L8-BO with different L8-BO ratios are shown in Fig. 2c. It can be observed that device based on D18:S-Cb exhibits a higher EQE response than the device based on D18:L8-BO in the ranges of 300-540 nm and 620-950 nm. Additionally, as the proportion of L8-BO increases from 25% to 75%, the EQE response intensity of D18:S-Cb:L8-BO first increases and then decreases. The optimal EQE spectrum response occurs when 50% of L8-BO is added, with improvements observed in 380-950 nm wavelength range. When the proportion of L8-BO increases to 75%, the EQE response is between that of D18:S-Cb and D18:L8-BO. Finally, the EQE-integrated *J*$_{SC}$ for D18:S-Cb:L8-BO-based OSCs with 0%, 25%, 50%, 75%, and 100% L8-BO contents are 26.56, 26.85, 27.09, 26.29, and 25.78 mA cm$^{-2}$, respectively, which agrees well with measured *J*$_{SC}$ with a reasonable error of approximately 4%.

**Carrier behavior and energy loss**

Femtosecond-resolved transient absorption spectra (fs-TAS) were measured for D18:S-Cb:L8-BO active layer with varying L8-BO ratios to probe photoinduced carrier dynamics in OSC devices. Spectral cuts at representative pump-probe delay times and corresponding 2D TAS contour plots are shown in Supplementary Fig. 15 and Supplementary Fig. 16, respectively. Donor polaron kinetics were fitted using the sum of exponentials, as shown in Fig. 2d. The kinetics of donor polarons photobleaching (PB) were used as the primary basis for analyzing free charge generation and sub-nanosecond recombination. It is clearly observed that the D18:L8-BO active layer facilitates very rapid singlet exciton dissociation, whereas this kinetic process slows down significantly in the D18:S-Cb blend. Conversely, the rapid dissociation process in D18:L8-BO may lead to an excessive concentration of electron-hole pairs that results in charge accumulation and promotes

free charge recombination. In contrast, the charge recombination process in D18:S-Cb slows down, as indicated by its more gradual decay trend (green curve, 0% L8-BO), while the D18:L8-BO exhibits a faster decay after 100 ps (red curve, 100% L8-BO), which is beneficial for achieving higher EQE and FF in devices. When coming into the ternary blend, the singlet exciton dissociation dynamics are further slowed, particularly when the proportion of L8-BO increases to 50% and 75%. With a 25% L8-BO ratio, free charge recombination in D18:S-Cb:L8-BO ternary blend will be slightly suppressed compared with no addition. The recombination process can be mostly inhibited when the L8-BO proportion is increased to 50%, which correlates well with enhanced EQE and FF in ternary OSCs based on D18:S-Cb:L8-BO with 50% L8-BO content. However, when the L8-BO ratio increases to 75%, free charge recombination in D18:S-Cb:L8-BO blend will increase, leading to a decrease in both *J*$_{SC}$ and FF for the corresponding device.

Carrier mobilities of optimal D18:S-Cb:L8-BO active layer with varying L8-BO contents were measured with resulting *J*-*V* curves displayed in Supplementary Fig. 17. Corresponding hole and electron mobilities ($\mu_h$ and $\mu_e$) fitted using SCLC method are summarized in Fig. 2e. D18:S-Cb active layer exhibits hole and electron mobilities of $6.45 \times 10^{-4}$ and $5.96 \times 10^{-4}$ cm$^2$ V$^{-1}$ s$^{-1}$, respectively, higher than those of D18:L8-BO ($3.49 \times 10^{-4}$ and $2.19 \times 10^{-4}$ cm$^2$ V$^{-1}$ s$^{-1}$, respectively). It should be noted that the high and balanced carrier mobility of D18:S-Cb helps reduce charge recombination, resulting in a lower recombination rate compared to D18:L8-BO. When L8-BO is introduced, both the hole and electron mobilities of D18:S-Cb:L8-BO decrease to varying extents relative to those of D18:S-Cb. D18:S-Cb:L8-BO with 50% L8-BO content exhibits the highest hole and electron mobilities with the lowest $\mu_h/\mu_e$ ratio in the ternary blend, which helps further reduce charge recombination (consistent with the fs-TAS test results) to promote improvements in *J*$_{SC}$ and FF.

The Fourier transform photonic spectroscopy EQE (Supplementary Fig. 18) and electroluminescence (EL) spectra (Fig. 2f) of D18:S-Cb:L8-BO system were measured to assess *E*$_{loss}$ of OSCs based on Shockley-Queisser (S-Q) theoretical framework. The calculated

**Table 1 | Photovoltaic parameters of optimized D18:S-Cb:L8-BO devices with varying L8-BO ratios**

| L8-BO ratio | $V_{oc}$ (V) | $J_{sc}$ (mA cm$^{-2}$) | $J_{sc}$[a] (mA cm$^{-2}$) | FF (%) | PCE[b] (%) |
|---|---|---|---|---|---|
| 0% | 0.888 | 27.57 | 26.56 | 80.2 | 19.63 (19.37 ± 0.24) |
| 25% | 0.901 | 27.97 | 26.85 | 81.1 | 20.44 (20.01 ± 0.23) |
| 50% | 0.907 | 28.21 | 27.09 | 81.8 | 20.93 (20.60 ± 0.16) |
| 75% | 0.916 | 27.39 | 26.29 | 81.3 | 20.40 (19.97 ± 0.25) |
| 100% | 0.925 | 26.83 | 25.78 | 80.0 | 19.85 (19.53 ± 0.27) |
| 50%[c] | 0.913 | 27.90 | - | 81.4 | 20.74 |

[a]The integrated $J_{SC}$ is calculated from the EQE spectra.

[b]Values in parentheses represent the average and standard deviation from 10-15 individual devices.

[c]The certified efficiency was obtained from the South China National Center of Metrology, Guangdong Institute of Metrology.

radiative and non-radiative $E_{loss}$[49] data are summarized in Supplementary Table 3. For D18:S-Cb:L8-BO blend with 0%, 25%, 50%, 75%, and 100% L8-BO content, the measured EQE$_{EL}$ values were determined to be $1.93 \times 10^{-4}$, $2.35 \times 10^{-4}$, $3.12 \times 10^{-4}$, $1.51 \times 10^{-4}$, and $1.45 \times 10^{-4}$, respectively, corresponding to non-radiative $E_{loss}$ ($\Delta E_3$) of 0.222, 0.217, 0.210, 0.229, and 0.230 eV, respectively, according to $\Delta E_3 = -kT\ln(EQE_{EL})$, where $k$ is the Boltzmann constant, and $T$ is Kelvin temperature. The slightly higher EQE$_{EL}$ and reduced $\Delta E_3$ of D18:S-Cb-based OSCs are likely attributed to suppressed electron-phonon coupling. It can be observed that L8-BO addition further enhances EQE$_{EL}$ of the ternary active layer, especially when L8-BO content reaches half of the total acceptor amount, where the EQE$_{EL}$ significantly increases, and $\Delta E_3$ is further suppressed. However, when L8-BO content is increased to 75%, the EQE$_{EL}$ of D18:S-Cb:L8-BO decreases, indicating an increase in $\Delta E_3$ (the reason will be discussed in the theoretical calculation section). $E_{loss}$s of D18:S-Cb:L8-BO-based OSCs with 0%, 25%, 50%, 75%, and 100% L8-BO content were calculated to be 0.523, 0.517, 0.515, 0.512, and 0.520 eV, respectively. It should be noted that D18:S-Cb-based OSCs obtain a slightly larger $E_{loss}$ than that of D18:L8-BO-based OSCs due to a larger additional radiative $E_{loss}$ ($\Delta E_2$).

### Working mechanism of the ternary device

As L8-BO content gradually increases, the $V_{OC}$ of D18:S-Cb:L8-BO-based ternary devices exhibits a linear improvement (Supplementary Fig. 19), suggesting that the potential mechanism for improved performance of ternary devices may be the formation of an acceptor alloy phase.

To thermodynamically probe the ease of forming an acceptor alloy phase between S-Cb and L8-BO, we evaluated their compatibility by measuring the contact angles of their neat films and deriving the corresponding surface tensions. By using ethylene glycol (EG) and water as probe liquids (contact angles of 65.40°/94.19° for L8-BO and 61.66°/89.77° for S-Cb, respectively, Supplementary Fig. 20), the surface tensions ($\lambda$) of L8-BO ($\lambda_{L8-BO}$) and S-Cb ($\lambda_{S-Cb}$) were fitted via Wu's model to be 39.68 mN/m and 40.33 mN/m, respectively. The Flory-Huggins interaction parameter $\chi$, estimated from $\chi \propto (\sqrt{\lambda_{S-Cb}} - \sqrt{\lambda_{L8-BO}})^2$ was found to be exceptionally small ($\chi \propto 0.0026$). This minimal $\chi$ value indicates excellent thermodynamic miscibility between S-Cb and L8-BO, providing a key thermodynamic rationale for their facile formation of a homogeneous acceptor alloy phase in the blend.

To further validate the working mechanism of ternary devices, we investigated the trend of energy levels in S-Cb:L8-BO blend films as L8-BO content varied by testing their CV curves (Supplementary Fig. 21). It was found that as L8-BO ratio increased, the LUMO energy levels of S-Cb:L8-BO blend exhibited a linear increase, while the HOMO energy levels showed the opposite trend. We also tested the DSC curves of S-Cb:L8-BO mixture (Supplementary Fig. 22), where only one single crystallization peak can be observed as L8-BO content gradually increased, indicating that S-Cb can form a good alloy phase with any ratio of L8-BO. However, as L8-BO content increased from 25% to 75%,

the crystallization temperature gradually decreased, and the enthalpy change of the corresponding crystallization peak significantly lowered, suggesting that an increase in L8-BO content will weaken the crystallization performance of the acceptor alloy phase. These two experiments' results revealed that the operating mechanism of the ternary device involves the formation of an S-Cb:L8-BO acceptor alloy phase.

### Single crystal analysis and theoretical simulations

To deeply understand the formation mechanism of S-Cb:L8-BO alloy phase at a molecular scale, a single crystal of S-Cb was analyzed, and theoretical simulations were employed to visualize intermolecular interactions between S-Cb and L8-BO. The single crystal of S-Cb was successfully grown using a three-solvent diffusion method with chlorobenzene as the good solvent, $n$-hexane as the poor solvent, and toluene as the buffer solvent. The crystallographic parameters of S-Cb are summarized in Supplementary Table 4. The results of the single crystal are shown in Fig. 3a-d. We found a Y-type dimer packing with overlapping molecular backbones in S-Cb (Fig. 3e). The π-π interaction distance between dimer molecules is about 3.31 Å. Similarly, we analyzed three types of dimer stacking from the L8-BO single crystal, namely S-type, S*-type, and M-type dimer packing with overlapping molecular backbone (Fig. 3f-h). The difference between S-type and S*-type is that S* only overlaps at the end groups. The π-π stacking distances of these types are 3.21 Å, 3.32 Å, and 3.27 Å, respectively, which are common stacking types and π-π distances in single crystals of Y6-series SMAs.

Based on quantum chemical calculations, the weak interaction analysis of dimers found that the overlapping visual in green color had an obvious π-π interaction, as shown in Fig. 3i-l. Overall, regardless of the type of molecular stacking found in the single crystal growth of S-Cb or L8-BO, we use δG (See Supplementary Information for its definition) to quantitatively describe the interaction strength of the four types of stacking (Y, S, S* and M), which are 1.59, 1.43, 1.32 and 1.46 respectively, that is, these types of molecular packing have π-π interactions with similar strengths. Further, the excited state theoretical calculation analysis of these dimer types shows that the distance of electron-hole separation $d_{e-h}$ in the first excited state is 18.05 Å, 17.34 Å, 19.88 Å and 17.86 Å, respectively, as shown in Fig. 3m-p. The difference in electron-hole separation, on the one hand, depends on the type of molecular packing, and on the other hand, it is determined by Coulomb attraction energy ($E_c$)[50]. For weak-bound carriers dominated by π-π weak interactions, $E_c$ is determined by the following Eq. (1):

$$E_c = \int \int \frac{\rho^{hole}(\mathbf{r_1})\rho^{ele}(\mathbf{r_2})}{|\mathbf{r_1} - \mathbf{r_2}|} d_{r1} d_{r2} \tag{1}$$

Where $\rho^{hole}$ and $\rho^{ele}$ are the density of holes and electrons, respectively, and $|r_1 - r_2|$ is the electrons-holes separation distance. That is, the Coulomb attraction energy and the separation distance between electrons and holes $d_{e-h}$ are inversely proportional. Our theoretical calculation results show that the Coulomb attraction energies of the

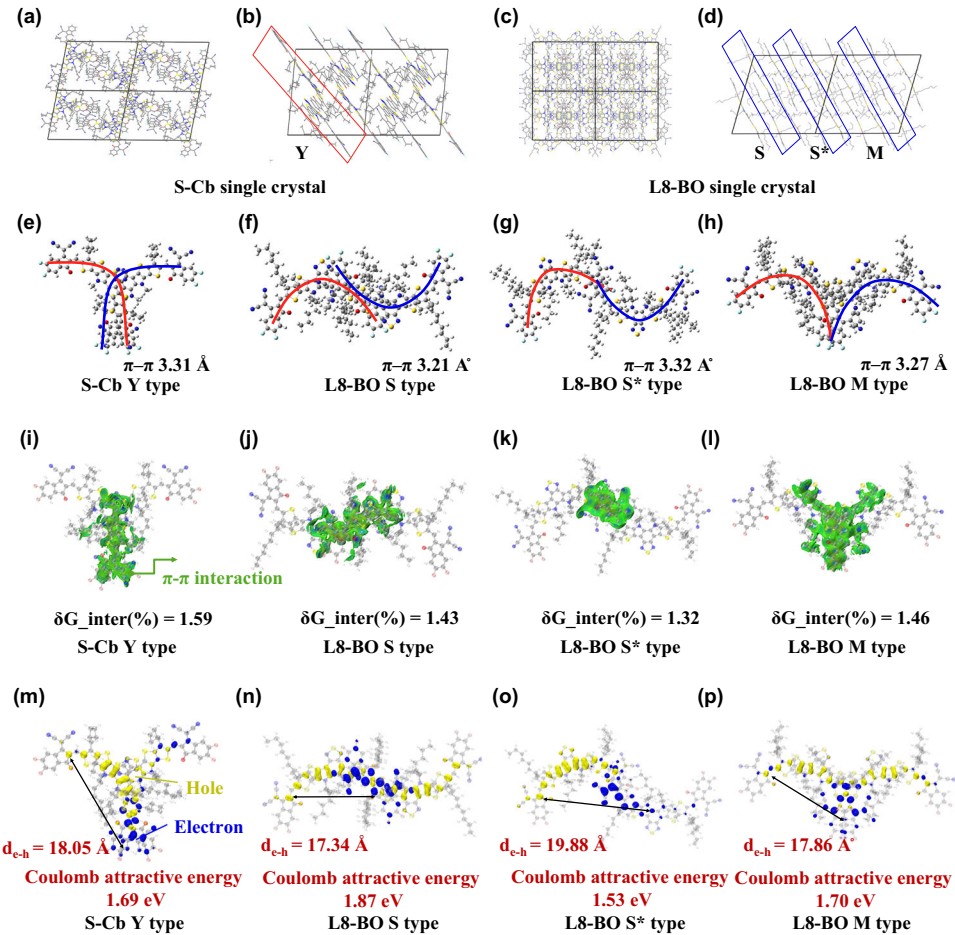

**Fig. 3 | Single crystal and quantum chemical calculation. a–d** Single crystal of S-Cb and L8-BO. Dimer stacking type in single crystal: **e** Y type, **f** S type, **g** S* type, and (**h**) M type. **i–l** Visualized π-π weak interaction. **m–p** Coulomb attractive energy and electron-hole separation distance.

four stacking types are very close, namely 1.69, 1.87, 1.53, and 1.70 eV, respectively. In summary, the four stacking types obtained by S-Cb and L8-BO single crystal have very close π-π stacking distances, π-π interaction strengths, and Coulomb attraction energies, as well as the resulting charge separation in the excited state.

Interestingly, we used molecular dynamics simulations to change different ratios of L8-BO in D18:S-Cb:L8-BO to simulate the formation of real blend films (Supplementary Fig. 23), and we found a unique dimer formed between S-Cb and L8-BO. As shown in Fig. 4a, we controlled the ratio of S-Cb:L8-BO to be 0.75:0.25 (dark blue), 0.50:0.50 (dark red), and 0.25:0.75 (light blue), and statistically analyzed the pair distribution $g_{S\text{-}Cb\text{-}L8\text{-}BO}$ with different dimer packing distances. According to theoretical calculation results, when the two ratios of acceptors are very different, that is, the ratio is 0.75:0.25 (dark blue) and 0.25:0.75 (light blue), the dimer packing distances of the two are concentrated in the range of 2.75- 4.31 Å. This stacking distance usually means that the two acceptors exist in a way that the molecular skeleton overlaps up and down, which is like the four types in single crystals mentioned above. More importantly, when the ratio of the two acceptors is 0.50:0.50 (dark red), a new dimer pair distribution peak is obtained in the stacking distance range of 1.56 to 2.73 Å, which means that S-Cb and L8-BO interact together more compactly.

To further explore this new stacking mode, we extracted all dimers of S-Cb and L8-BO with stacking distances ranging from 1.56 to 2.73 Å from the simulation results of molecular dynamics. As shown in Fig. 4d, the nearly planar cyclobutyl side chains of S-Cb are clamped in the dendritic side chains of L8-BO. This incredible intermolecular

stacking mode is achieved through hydrogen bonds on the molecular side chains. The dark blue part in Fig. 4d shows this attraction between hydrogens. Compared with the π-π interaction, the hydrogen bond makes the two molecules more firmly clamped together. Compared with the four stacking modes in the single crystal, the δG of the clamp type is only 0.65, and the clamp distance between them is only in the range of 2.23 to 2.96 Å. By counting the S-Cb and L8-BO stacking types at different ratios, as shown in Fig. 4b, we found that when the ratio of the two acceptors is 0.50:0.50 (center), the clamp effect (refers to a supramolecular interaction wherein the nearly planar and rigid cyclobutyl side chain of S-Cb is spatially locked into the branched alkyl chain pocket of L8-BO via intermolecular hydrogen bonds, e.g., C-H···H-C, effectively clamping the side chains of the two acceptors together) is most obvious, and the clamp type ratio reaches 27%; when the ratios of the two acceptors are different, 0.75:0.25 (left) and 0.25:0.75 (right), the clamp effect still exists but is weakened to varying degrees, depending on the content of S-Cb, 16% and 8%, respectively. More interestingly, we also counted the impact of the clamp effect on the packing of acceptors and donors. The results show that when the ratio of the two acceptors is the same (0.50:0.50), the stacking of donors and acceptors is the largest, at 28%; and when the ratios of the two acceptors are different, the stacking of donors and acceptors will also be affected, at 20% and 17% respectively, and the influence may also be related to the content of S-Cb. We believe that the reason for this phenomenon may be that the clamp effect caused by mixing two different acceptors originates from the difference in the side chains of the two molecules. The nearly planar alkyl chain of S-Cb is more likely to be

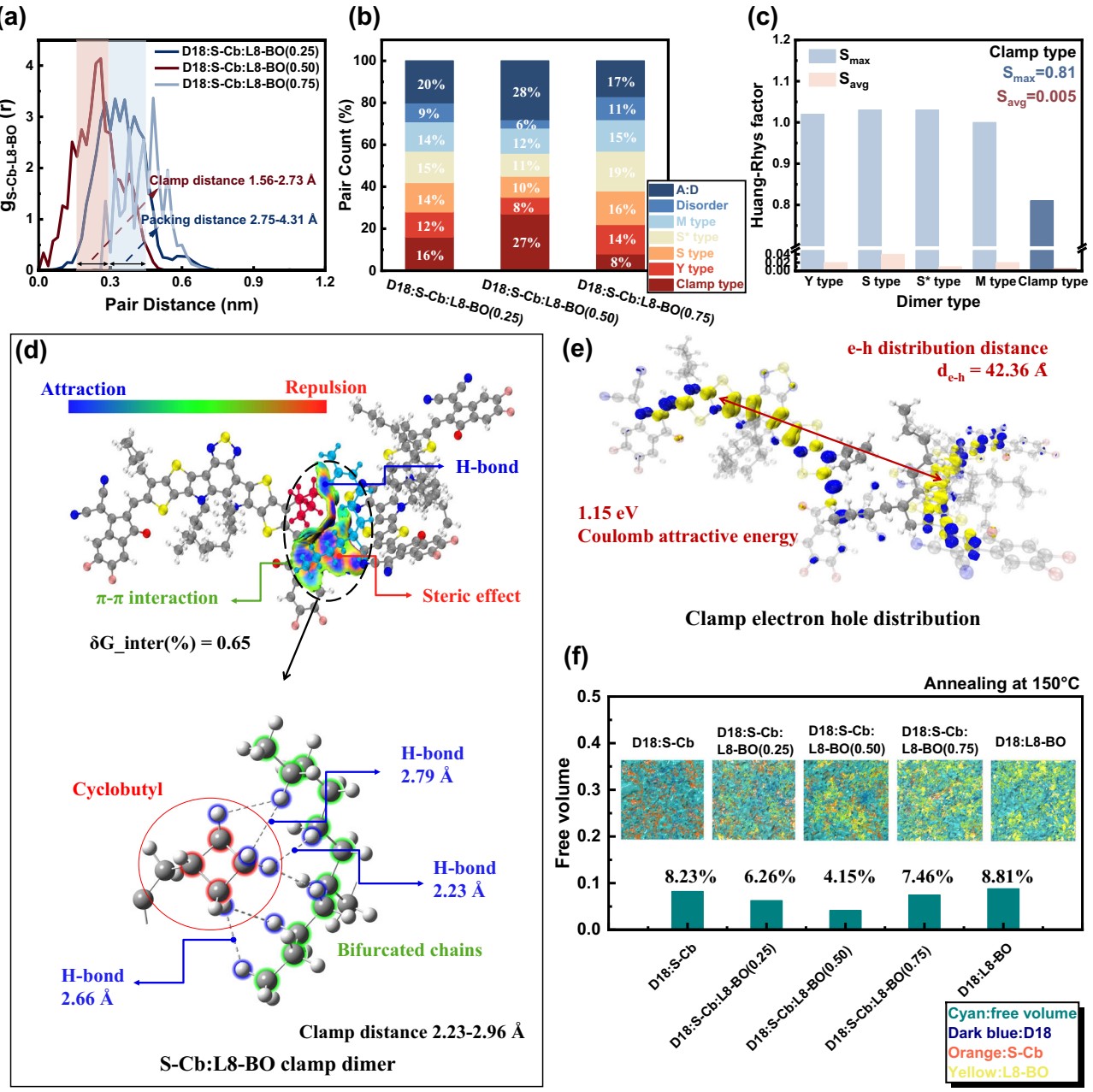

**Fig. 4 | Molecular dynamics simulations. a** Dimer pair distance (r) as a function of pair distribution $g_{S\text{-}Cb\text{-}L8\text{-}BO}(r)$. **b** Dimer types pair counts in different L8-BO ratios in D18:S-Cb:L8-BO ternary blend. **c** Huang Rhys factor of dimer types. **d** Weak interaction in a clamp-type dimer. **e** Coulomb attractive energy and electron-hole separation distance in a clamp-type dimer. **f** Simulated free volume and annealing donors and acceptors blend at 150 °C.

captured by the dendritic side chains than the linear alkyl chain, and the probability of being clamped increases to a certain extent with the increase of S-Cb, reaching the maximum when the ratio is 0.50:0.50; the clamp dimer is combined by hydrogen bonds, and does not affect the further stacking of the molecular backbone with other molecules, that is, the molecules form a fiber-like chain, and the clamp molecular chain still can be arranged with other molecules (including donors) through backbone stacking. This is also the reason why the stacking probability of the donor and the acceptor increases when the ratio of the two different acceptors is 0.50:0.50.

Compared with the four stacking types discussed previously, we found through theoretical calculations that the clamp dimer has a smaller Coulomb attraction of only 1.15 eV, which greatly enhances the electron-hole separation distance to 42.36 Å, which means that

electrons and holes can be more effectively separated at the interface between the acceptor pure domain and blend interface (Fig. 4e). At the same time, we compared the effects of the stacking types in single crystals and the clamp type on the Huang Rhys factor, and Eq. (2) is:

$$S = \frac{\lambda}{\hbar\omega} \tag{2}$$

Where $\lambda$ is the reorganisation energy and $\omega$ is the molecular vibration frequency. The maximum values of the Huang Rhys factor ($S_{max}$) of the four dimer types are 1.02, 1.03, 1.03, and 1.00, and the average values ($S_{avg}$) are 0.02, 0.04, 0.01, and 0.02, respectively. Correspondingly, the $S_{max}$ of the clamp-type dimer is 0.81, and the $S_{avg}$ is 0.005 (Fig. 4c and Supplementary Fig. 24). This may be because the clamp stacking relies

on hydrogen bonds between side chains to suppress the high-frequency vibration[51] and weaken the reorganization energy of the low-frequency vibration mode[52,53]. In addition, a lower free volume was found in the ratio of 0.50:0.50 (Fig. 4f). The clamping effect helps to reduce both the Huang-Rhys factor and the free volume of the ternary active layer, thereby effectively suppressing electron-phonon coupling to improve $EQE_{EL}$ of D18:S-Cb:L8-BO ternary active blend and suppress $\Delta E_3$[21].

To quantitatively evaluate the electron-phonon coupling strength, temperature-dependent PL measurements were performed on L8-BO, S-Cb neat films, and their blended film with a 1:1 weight ratio. The temperature was varied from 300 K down to 80 K in steps of 20 K. The PL spectra and corresponding fitting results are provided in Supplementary Fig. 25. The broadening of the PL emission linewidth with increasing temperature can generally be described by the following model[19,54,55]:

$$\Gamma(T) = \Gamma_i + a\,exp\left(-\frac{Ea}{k_b T}\right) + bT \tag{3}$$

where, $\Gamma_i$ represents the inhomogeneous linewidth of the film, $a$ denotes the density of non-radiative recombination centers, $E_a$ is regarded as the energy barrier for back charge recombination from electronic to excitonic states in Y-series non-fullerene acceptors, and b is defined as the electron-phonon coupling coefficient. In the low-temperature region (80-200 K), the contribution of the second (nonlinear) term to the linewidth is minor; therefore, an approximate linear fitting was applied to the data in this range to extract the electron-phonon coupling constant b. The fitting yields b values of approximately 0.305, 0.284, and 0.163 meV·K$^{-1}$ for neat L8-BO, S-Cb, and the S-Cb:L8-BO blend, respectively. These results indicate that the electron-phonon coupling in neat S-Cb is slightly weaker than that in neat L8-BO. Notably, blending the two materials at a 1:1 weight ratio leads to a more pronounced suppression of the coupling, a trend consistent with our theoretical calculations. We attribute this enhanced suppression to the formation of larger dimer-like structures via hydrogen-bonding interactions between the side chains of S-Cb and L8-BO, which likely further restrains molecular vibrations.

## H-H interaction of the clamping effect

To verify the inter-H interaction between cyclobutyl side chain of S-Cb and 2-butyloctyl side chain of L8-BO, which is observed in theoretical calculations, we performed $^1H$-$^1H$ NOESY NMR spectroscopy on S-Cb, L8-BO, and S-Cb:L8-BO (1:1) mixture (Supplementary Fig. 26). The hydrogen atoms labeling Ha-Hh and their positions on S-Cb and L8-BO are marked in Supplementary Fig. 26a. The $^1H$-$^1H$ NOESY NMR spectrum of S-Cb (Supplementary Fig. 26b) shows long-range H-H interactions between Ha and Hb, Hd and Hc, and Hd and Hb on cyclobutyl side chain. In contrast, no long-range H-H correlation is observed between Hf and Hg in L8-BO (Supplementary Fig. 26c), likely due to the flexibility of branched alkyl chains, which causes Hf and Hg to be spatially distant. It is worth noting that a strong H-H spatial interaction exists between Hf and Hh, whereas such an interaction does not occur between Ha and He in S-Cb due to greater spatial distance between Ha and He in S-Cb than that between Hf and Hh in L8-BO as revealed by single crystal analysis (Supplementary Fig. 27a). This phenomenon reveals that cyclization strategy can reduce the steric hindrance acting on the terminal double bonds to promote the planarity of S-Cb, which is consistent with the single crystal results (Supplementary Fig. 27b). As can be observed in $^1H$-$^1H$ NOESY NMR spectrum of S-Cb:L8-BO blend solution with equal ratio (Supplementary Fig. 26d), in addition to the H-H correlations within the individual molecules of S-Cb and L8-BO, new intermolecular long-range H-H correlations appeared between Hb and Hg, and Ha and Hf, which indicates that Hb and Hg, and Ha and Hf, are spatially very close. Furthermore, the long-range H-H correlation

intensity between Ha and Hf is weaker than that between Hb and Hg, suggesting that the rigid cyclobutyl side chain of S-Cb can effectively clamp the branched alkyl chain at the branching point of L8-BO, which is in agreement with the molecular dynamics results.

To verify the generality of the clamping effect induced by the cyclobutyl group, specifically probing whether effective long-range interactions exist with the linear alkyl chains on the acceptor molecule, we performed $^1H$-$^1H$ NOESY NMR spectroscopy on a 1:1 blend of S-Cb and BTP-eC9. The relevant hydrogen atoms on BTP-eC9 are labeled as Hi-Hk, as illustrated in Supplementary Fig. 26a. The NOESY spectrum reveals distinct spatial proximity correlations between the hydrogen atoms of the cyclobutyl group on S-Cb (Ha) and those on the linear alkyl chains of BTP-eC9 (Hj and Hi) (Supplementary Fig. 26e). The intensity of this interaction appears stronger than that previously observed in the S-Cb:L8-BO system, which can be attributed to the lower steric hindrance of linear alkyl chains compared to branched ones that allows the cyclobutyl group to approach the main body of the linear chain more closely. This result provides evidence for a favorable spatial proximity and orientation between the cyclobutyl group of S-Cb and the linear alkyl chain-based acceptor, forming a crucial structural foundation for the clamping effect. This finding further reinforces that the cyclobutyl-induced clamping effect of S-Cb may a general phenomenon applicable to acceptor molecules functionalized with either branched or linear alkyl side chains.

To further prove the existence of C-H…H-C hydrogen bond, we have performed the Fourier-transform infrared (FTIR) spectroscopy on neat films of S-Cb and L8-BO, as well as their blend film with a weight ratio of 1:1, corresponding spectra are provided in Supplementary Fig. 28. By comparing the FTIR transmission spectra of the neat S-Cb and L8-BO films, we assigned the characteristic C-H stretching vibration of the cyclobutyl group at approximately 2954 cm$^{-1}$. Notably, in the S-Cb:L8-BO blend film, this C-H vibration peak exhibits a small but discernible shift to a higher wavenumber (blue-shift of about 2 cm$^{-1}$) compared to the neat S-Cb film, accompanied by an increase in peak intensity and broadening of the line shape. These spectral changes align with the trend expected when the cyclobutyl C-H bond acts as a potential hydrogen-bond acceptor. It is worth noting that the considerable strain and the resulting bent bonds enhance the polarity of the C-H bonds on cyclobutyl, endowing the hydrogen atoms with a partial positive charge, though still weaker than that of hydrogens bonded directly to oxygen or nitrogen. The compact steric profile and the weakly electrophilic hydrogen of the cyclobutyl group allow it to act as a hydrogen-bond acceptor, forming weak but non-negligible hydrogen bonds. We consider that the relatively small magnitude of the shift may be attributed to the moderate polarity of the cyclobutyl C-H bond, which could limit the strength of its hydrogen-bond accepting interaction, thereby resulting in only subtle yet detectable alterations in the FTIR spectrum.

## Clamping effect on ternary morphology

Intermolecular stacking properties of D18:S-Cb:L8-BO active layer with varying L8-BO contents were explored through GIWAXS experiment, and the resulting 2D GIWAXS patterns and corresponding 1D cutting line profiles are shown in Supplementary Fig. 29. It reveals that all D18:S-Cb:L8-BO blend films exhibit a well-defined face-on orientation, which facilitates the vertical transport of charge carriers. The π-π stacking distances/CLs for D18:S-Cb and D18:L8-BO blend films are 3.742/15.09 Å and 3.683/13.80 Å, respectively. Although the π-π stacking distance of D18:S-Cb is slightly larger than that of D18:L8-BO, the cyclobutyl substitution is able to enhance CL of D18:S-Cb blend film, which is consistent with pure film measurements of S-Cb and L8-BO and helps explain a higher carrier mobility of D18:S-Cb than that of D18:L8-BO. As L8-BO content increases to 25%, 50%, and 75%, the π-π stacking distances/CLs of D18:S-Cb:L8-BO active layer are found to be 3.642/15.42 Å, 3.611/16.97 Å, and 3.630/14.90 Å, respectively. This

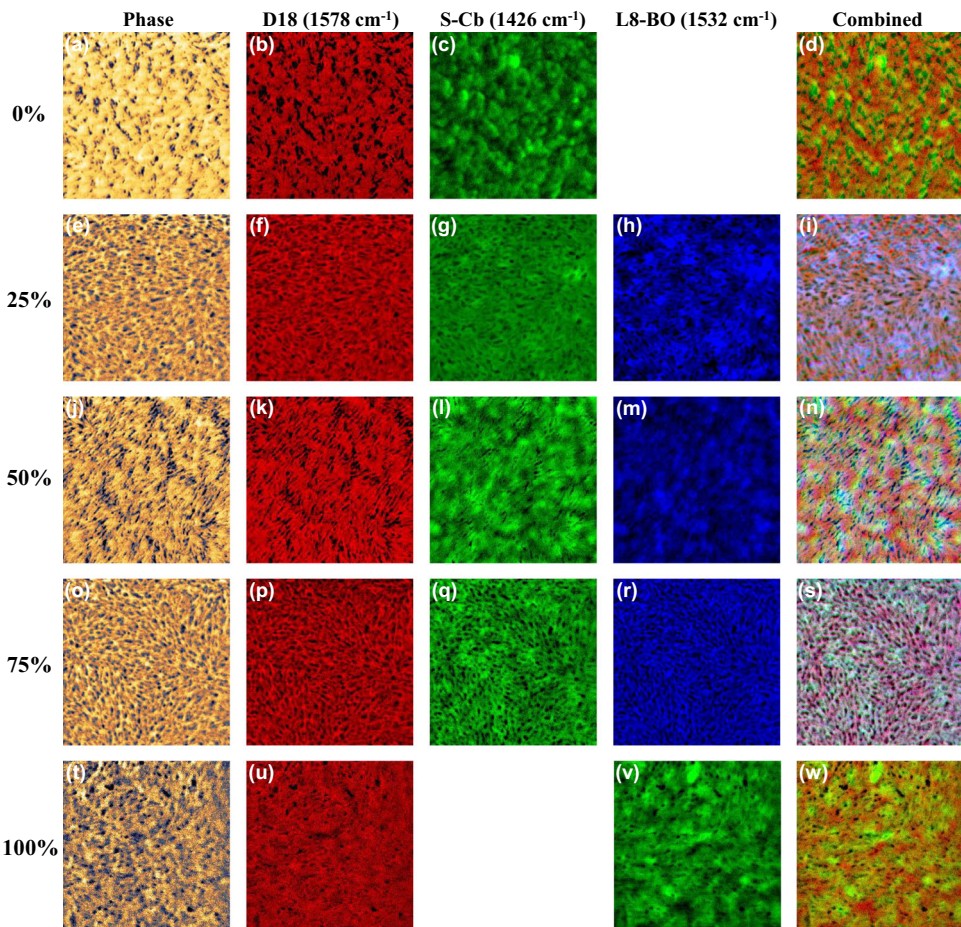

|  | Phase | D18 (1578 cm⁻¹) | S-Cb (1426 cm⁻¹) | L8-BO (1532 cm⁻¹) | Combined |

**Fig. 5 | Morphological characterization.** PiFM images of the blend films: **a–d** D18:S-Cb; **e–i** D18:S-Cb:L8-BO (25 wt% L8-BO); **j–n** D18:S-Cb:L8-BO (50 wt% L8-BO); **o–s** D18:S-Cb:L8-BO (75 wt% L8-BO); **t–w** D18:L8-BO.

trend supports the variation in carrier mobility of D18:S-Cb:L8-BO blend films. It can be concluded that the clamping effect between S-Cb and L8-BO plays a crucial role in regulating the morphology of the ternary active layer. When S-Cb and L8-BO are in equal proportions, the clamping effect between S-Cb and L8-BO reaches its maximum, significantly reducing the π-π stacking distance and enhancing CLs of the active layer. However, when the ratio of S-Cb to L8-BO is unequal, especially when L8-BO is in excess, the clamping effect is lowered, which weakens the π-π stacking. GIWAXS results indicate that the clamping effect between S-Cb and L8-BO is beneficial for improving π-π stacking properties in the ternary active layer.

To further understand impacts of clamping effect between S-Cb and L8-BO on the morphology of ternary active layer, photo-induced force microscopy (PiFM) was employed to image the characteristic Fourier-transform infrared (FTIR) absorption peaks of the polymer donor (1578 cm⁻¹ for D18) and two acceptors (1426 cm⁻¹ for S-Cb and 1532 cm⁻¹ for L8-BO, Supplementary Fig. 30). PiFM images of D18:S-Cb:L8-BO with 0%, 25%, 50%, 75%, and 100% L8-BO ratios are shown in Fig. 5. By comparing the phase diagrams of D18:S-Cb and D18:L8-BO blend films, as presented in Fig. 5a, t, significant morphological differences between D18:S-Cb and D18:L8-BO blend films can be clearly observed. The D18:S-Cb blend exhibits a hilly surface feature, while the D18:L8-BO blend shows a smooth fish-scale morphology. Figure 5c displays the distribution morphology of S-Cb (green) acceptor component in D18:S-Cb, which reveals a peak-like morphology compared to that of L8-BO acceptor component (Fig. 5v) in D18:L8-BO, which can be attributed to the rigidity of cyclobutyl that effectively improves molecular crystallization properties of S-Cb. Figure 5b shows the

distribution morphology of polymer donor D18 (red) in the D18:S-Cb component, which corresponds to the portion remaining after S-Cb is removed from D18:S-Cb blend. D18 component in D18:S-Cb showed a more obvious pit structure, which in turn indicated that S-Cb could form a better pure acceptor phase. The phase length scales of D18 and S-Cb components in D18:S-Cb blend are comparable to those in D18:L8-BO and are approximately 15 nm and 35 nm, respectively (Supplementary Fig. 31), which favors exciton transfer to donor/acceptor interfaces before quenching. Additionally, the PiFM combined images of D18:S-Cb blend (Fig. 5d) show a more significant donor-acceptor phase separation compared to D18:L8-BO. The stronger crystallinity of S-Cb helps S-Cb acceptor phase to precipitate faster, facilitating the formation of a more continuous and high-quality donor-acceptor nanofiber network structure and carrier transport channel, which enhances electron and hole mobilities. In contrast, the PiFM combined images of D18:L8-BO show more light yellow areas, indicating a higher degree of donor-acceptor blending phase. Overall, the cyclobutyl cyclization strategy increases the donor-acceptor phase separation by enhancing molecular crystallinity.

When L8-BO is introduced with a ratio of 25% and 75%, the phase diagrams of D18:S-Cb:L8-BO active layers show a leopard-like surface morphology (Fig. 5e, o), which is likely due that the clamping effect between S-Cb and L8-BO is not strong enough when the mass ratios of S-Cb to L8-BO are 3:1 and 1:3, preventing forming a high-quality acceptor alloy phase between S-Cb and L8-BO. From PiFM images of S-Cb and L8-BO components in D18:S-Cb:L8-BO blend (Fig. 5g, h, q, r), clear crystalline phases and nanofiber network structures of S-Cb and L8-BO can be observed, indicating more pure phases of S-Cb and L8-

BO components in D18:S-Cb:L8-BO ternary blend with L8-BO content of 25% and 75%, particularly evident when the L8-BO content reaches 75%. However, when the ratio of S-Cb to L8-BO is the same, the clamping effect between the two acceptors is maximized, and S-Cb and L8-BO are able to form a high-quality acceptor alloy. As shown in Fig. 5i, m, the crystalline phases and nanofiber network structures of S-Cb and L8-BO components become less prominent, which is significantly different from the morphology of S-Cb and L8-BO components observed in D18:S-Cb:L8-BO blend with 25% or 75% L8-BO content. The uniform and smooth microstructure of S-Cb and L8-BO components in D18:S-Cb:L8-BO indicating that the two are more inclined to be completely intermixed. Interestingly, the addition of L8-BO significantly increases the phase length scale of D18 from 15 nm to ~25 nm, while reducing the phase length scale of acceptor phases from 35 nm to ~25 nm (Supplementary Fig. 31). This appropriately matched phase length scale between donor and acceptor is considered to be beneficial for improving carrier transport and reducing charge recombination, consistent with test results above. Moreover, PiFM combined images of D18:S-Cb:L8-BO blend with 50% L8-BO content present a good donor-acceptor nano-interpenetrating network structure along with a more uniform donor-acceptor distribution compared to D18:S-Cb:L8-BO blend with 25% and 75% L8-BO content.

To further elucidate the role of clamping effect between S-Cb and L8-BO on morphology stability, PiFM measurements of degraded blend films under 100 °C for one day was conducted with results shown in Supplementary Fig. 32. It can be clearly observed from PiFM images (Supplementary Figs. 32a–d and 32j–m) that the degraded blend films of D18:S-Cb and D18:L8-BO underwent a great morphological change compared to previously fresh ones. The appearance of pore structures in both donor and acceptor components indicates severe phase aggregation. The phase length scales of D18, S-Cb, and L8-BO components in D18:S-Cb or D18:L8-BO blends all increase to ~ 60 nm. In contrast, the morphology of D18:S-Cb:L8-BO with 50% L8-BO content can be well maintained (Supplementary Figs. 32e–i), and the phase length scales of D18, S-Cb, and L8-BO are slightly increased to ~28 nm for donor and ~30 nm for acceptors, which benefitted from the clamping effect between S-Cb and L8-BO and significantly slow down the aggregation process of acceptor phases. These results informed us that the clamping effect between S-Cb and L8-BO plays a crucial role in helping to form high-quality acceptor alloy phases.

### Thermal- and photo-stability

To further investigate the impact of the clamping effect on thermal- and photo-stability of OSCs, we fabricated inverted devices with a structure of ITO/ZnO/active layer/MoO$_x$/Ag. The stability test results for devices based on D18:S-Cb:L8-BO with 0%, 50%, and 100% L8-BO content are presented in Supplementary Fig. 33. Under continuous annealing at 100 °C, D18:S-Cb device exhibited significantly better thermal stability than the D18:L8-BO device, which can be attributed to the higher $T_g$ of S-Cb resulting from its short and rigid cyclobutyl substituent. Furthermore, when 50% L8-BO was incorporated into the D18:S-Cb active layer, the device stability showed improvement within the first 160 h of heating. However, beyond 160 h, the thermal stability of the ternary device declined more rapidly compared to the D18:S-Cb binary device. We speculate that during the early stage of heating, the clamp effect mediated by hydrogen bonding helps to stabilize the morphology. With prolonged thermal stress, this hydrogen-bond network may gradually degrade, causing the ternary blend to evolve toward a morphology resembling that of D18:L8-BO, thus accelerating the decay in device performance.

It is worth noting that, under continuous heating at 100 °C, these weak interactions provide limited stabilization, which explains why the stability improvement is modest and eventually deteriorates over extended periods, leading to the observed trend of initial stability followed by rapid decay in the ternary system. In contrast, under continuous illumination, such hydrogen-bonding interactions appear less susceptible to degradation. Consequently, the device based on D18:S-Cb:L8-BO with 50% L8-BO content demonstrated markedly better photo-stability than the D18:S-Cb binary device. Although the overall thermal- and photo- stability of the ternary devices remains inferior to state-of-the-art values reported in the literature[56]—particularly in terms of thermal stability, these results provide indirect evidence supporting the role of the hydrogen-bond-guided clamping effect in modulating blend morphology and device stability.

## Discussion

In summary, an SMA named S-Cb, substituted with cyclobutyl groups at the thiophene $\beta$-position of the central core, was designed and synthesized. Leveraging the inherently high ring strain of cyclobutyl, the C-C bond vibrations within side chains can be slowed, and thus increasing the molecular rigidity of S-Cb, suppressing electron-phonon coupling in S-Cb. The molecular crystallization of S-Cb is superior to that of L8-BO, primarily due to enhanced lamellar packing. More importantly, the nearly planar structure of cyclobutyl allows for interchain supramolecular interactions between S-Cb and L8-BO via hydrogen bonding. This clamping effect reaches its maximum when the ratio of S-Cb to L8-BO is equal, which reduces the Huang-Rhys factor of acceptor dimers and lowers the FVR of ternary blends, thereby further suppressing electron-phonon coupling to enhance EQE$_{EL}$ of active layers. Moreover, this clamping effect between S-Cb and L8-BO also aids in the formation of a high-quality acceptor alloy phase, promoting balanced carrier transport and reducing charge recombination. Finally, ternary devices based on D18:S-Cb:L8-BO achieved a high efficiency approaching 21% along with a certified value of 20.74%. Overall, the cyclobutyl group promotes interchain supramolecular interactions between S-Cb and L8-BO, which not only suppresses electron-phonon coupling but also simultaneously improves ternary blend morphology for highly-efficient OSCs.

## Methods

### Device fabrications and characterizations

OSCs devices were fabricated by employing a conventional structure of ITO/PEDOT:PSS/D18/PFN-Br:MA/Ag. The patterned ITO-coated glass substrates were scrubbed with detergent, followed by ultrasonic treatment in deionized water, acetone, and isopropanol sequentially. The substrates were then dried overnight in an oven. Prior to use, the glass substrates were treated with UV-ozone for 30 minutes to enhance their work function and cleanliness. PEDOT:PSS (Al4083 from Heraeus) was spin-coated onto the ITO substrate at 7500 rpm for 30 seconds and subsequently dried at 160°C for 15 minutes under a nitrogen atmosphere. A blend solution of D18:S-Cb:L8-BO (D18 concentration: 4 mg/mL, with a weight ratio of D18 to acceptor at 1:1.4, and the ratio of L8-BO to acceptors varying from 0, 0.25, 0.50, 0.75 to 1.00) in chloroform, with 5 mg/mL 1-bromo-2,6-dichlorobenzene (DCBB) as an additive, was pre-dissolved at 100°C for 20 minutes. Once the solution cooled to approximately 60°C, it was spin-coated onto the PEDOT:PSS layer at 2000 rpm for 30 seconds and then annealed at 100°C for 1 minute. A thin layer of PFN-Br (0.5 mg/mL in methanol with 0.25% melamine by weight) was then spin-coated on the active layer at 3000 rpm. Subsequently, Ag was deposited by thermal evaporation at a pressure of 3 × 10$^{-4}$ Pa through a shadow mask. The optimal thickness of the active layer, measured with a Bruker Dektak XT stylus profilometer, was approximately 110 nm. The current density-voltage ($J$-$V$) characteristics of the devices were measured using a Keysight B2901A Source Meter inside a glove box under AM 1.5 G (100 mW cm$^{-2}$) illumination from an Enlitech solar simulator. The device contact area was 0.042 cm², and the illuminated area during testing was 0.040 cm², determined by a mask. External quantum efficiency (EQE) spectra were measured using a Solar Cell Spectral Response Measurement System QE-R3011 (Enlitech Co., Ltd.), with light intensity at each wavelength calibrated using a standard monocrystalline silicon photovoltaic cell.

## Theoretical simulations

Gaussian 16 (Revision C.02) code was used to perform density functional theory (DFT) calculations at the non-empirically tuned B3LYP-D3(BJ)/TZVP level of theory. The excited state single point energy was calculated under TD CAM-B3LYP/Def2TZVP level for high precision calculations. The wavefunction software Multiwfn and VMD were used for post-processing. Molecular dynamics simulations were conducted with the GROMACS 2023 software package. A spherical cut-off of 1.2 nm for the summation of van der Waals (VDW) interactions and short-range Coulomb interactions, and the particle-mesh Ewald solver for long-range Coulomb interactions were used throughout. The simulations were carried out with three-dimensional periodic boundary conditions using the leapfrog integrator with a time step of 1.0 fs. The electrostatic potentials were fitted using RESP charges. The velocity rescaling thermostat and Berendsen barostat under the NPT ensemble were applied to control the temperature and pressure, respectively.

## PiFM measurements

PiFM results were acquired using a VistaScope microscope from Molecular Vista inc. at Songshan Lake Materials Laboratory. All PiFM experiments were excited by a pulsed quantum cascade laser (Block Engineering) with a gap-free narrowband tunable wavenumber of 760-1950 cm$^{-1}$. The spectral linewidth is ~2 cm$^{-1}$ with a wavenumber resolution of 0.5 cm$^{-1}$. The PiFM experiment here was operated at the sideband excitation with the laser-frequency modulated at $f_m = f_1 - f_0$, where $f_0$ is the first mechanical eigenmode resonance of the cantilever that is used for PiF signal detection, while $f_1$ denotes the second ones recorded for the AFM topography of the sample. The probe is a Pt-coated tip with a resonant frequency of ~350 kHz (PPP-NCHPt-MB, Nanosensors).

## Reporting summary

Further information on research design is available in the Nature Portfolio Reporting Summary linked to this article.

## Data availability

All data supporting the findings of this study are available within the main text and the Supplementary Information file. Additional data are available from the corresponding author upon request. The X-ray crystallographic coordinates for the structures reported in this study have been deposited at the Cambridge Crystallographic Data Centre (CCDC) under deposition number 2423862 (S-Cb). The crystallographic data can also be obtained free of charge from the Cambridge Crystallographic Data Centre via www.ccdc.cam.ac.uk/data_request/cif.

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

## Acknowledgements

This work was supported by the National Natural Science Foundation of China under Grant Nos. U23A20371 (W.G.) and U21A2078 (Z.W.). Z.W. acknowledges the support from FuXiaQuan Self-created Zone Collaborative Project (3502ZCQXT2023006). W.G. acknowledges the support from the Scientific Research Funds of Huaqiao University (605-50Y23024) and the Xiamen Outstanding Young Talents Program (605-52424047). G.L. acknowledges the support from Research Grants Council of Hong Kong (Project Nos. 15221320, 15307922, and C4005-22Y), RGC Senior Research Fellowship Scheme (SRFS2223-5S01), the Hong Kong Polytechnic University: SirSze-yuen Chung Endowed Professorship Fund (8-8480), RISE (Q-CDC6), PRI (Q-CD7X, CDAJ), and Guangdong-Hong Kong-Macao Joint Laboratory for Photonic-Thermal-Electrical Energy Materials and Devices (GDSTC No. 2019B121205001). R.M. thanks the PolyU Distinguished Postdoctoral Fellowship (1-YW4C).

## Author contributions

W.G., H.X., R.M., Z.W., and G.L. conceived the idea. W.G. designed and synthesized S-Cb, and performed TGA, DSC, single-crystal growth and analysis, temperature-dependent NMR, and GIWAXS measurements. Y.H. and J.W. conducted the theoretical simulations. J.Z. carried out the UV-vis absorption, PL, CV, and SCLC measurements. R.M. and H.X. fabricated and characterized the device, including *J-V*, EQE, $E_{loss}$, certified efficiency, PiFM, and prepared samples for morphology testing. T.A.D.P. and J.W. performed the fs-TAS measurements and analyzed the data. W.G., H.X., R.M., Z.W., and G.L. contributed to the preparation of this manuscript, with C.D. and J.-X.T. providing constructive suggestions during the writing and revision of the first draft.

## Competing interests

The authors declare no competing interests.
