## [Transparent Peer Review file · Nature Communications]

Interchain supramolecular interactions drive nearly 21% efficiency organic solar cells

Corresponding Author: Professor Gang Li

Version 0:

Reviewer comments:

Reviewer #1

(Remarks to the Author)

Wei Gao and colleagues report organic solar cells achieving 20.93% efficiency (20.74% certified) through the incorporation of cyclobutyl-substituted acceptor S-Cb in ternary blends with D18 and L8-BO. They propose that these rigid cyclobutyl groups enable "clamping" interactions with L8-BO via hydrogen bonding, suppressing electron-phonon coupling and improving device performance. This manuscript reports a meaningful efficiency improvement in organic solar cells and employs comprehensive characterization techniques. However, the mechanistic claims, particularly regarding electron-phonon coupling suppression via the "clamping effect," are not definitively proven. The work represents a solid incremental advance through molecular engineering rather than a conceptual breakthrough. Despite this, I recommend publication subject to the following revisions:

Major Comments

- 1) The proposed mechanism linking cyclobutyl substitution to efficiency improvements involves multiple intermediate effects that are inadequately deconvoluted. The authors show changes in the molecular rigidity (T_g increase of 5.6°C), crystallinity (4× higher enthalpy), morphology (altered phase separation), and intermolecular interactions. However, the relative contributions remain unclear and improvements are automatically attributed to the clamping effect of cyclobutyl substitution. The authors need to provide a good statement on these intermediate effects and how much it contributes. For instance in line 198 the authors state 'Both S-Cb and L8-BO films exhibit well-defined face-on orientations, indicating that cyclobutyl substitution does not disrupt the preferential molecular orientation'. The emphasis on electron-phonon coupling suppression appears somewhat arbitrary given that morphological improvements alone could explain the performance gains.
- 2) The hydrogen bonding-mediated clamping between S-Cb and L8-BO feels overstated. MD simulations show the effect only at 50:50 ratio, raising questions about why this specific stoichiometry is special. Should we not see this in higher S-CB concentrations as well? Additionally, the effect of the 'unique dimer' has not been adequately explained.
- 3) The 1H-1H NOESY NMR provides only indirect evidence of proximity, not definitive proof of the proposed binding mode
- 4) I find that the correlation between computed Huang-Rhys factors and device performance is tenuous
- 5) While 10-15 devices were tested per condition, no statistical analyses determine whether performance differences are significant given the reported standard deviations (0.16-0.27%). The optimal 50:50 ratio could be within experimental error of other compositions.
- 6) The study examines only one system. Claims about the broader applicability of cyclobutyl-mediated interactions require demonstration in other donor-acceptor combinations. Should we expect the same behaviour with other binary and ternary blends as well? Or is the clamping effect only applicable to this particular combination.

Minor Comments

1. The single crystal analysis, while thorough, may have limited relevance to solution-processed films with substantial disorder.

2. PiFM characterization is sophisticated but interpretation of "acceptor alloy phase" from IR absorption alone is problematic.
3. Stability testing (100°C, 1 day) is insufficient for practical relevance. Light stability and longer-term studies are needed for a proper conclusion.
4. The Stokes shift reduction (2.3 nm) is marginal and within measurement uncertainty.
5. Fig. 4f showing free volume is difficult to interpret. Quantitative analysis would be preferable to color maps. I can't really see where the free volume is.
6. The manuscript would benefit from discussing synthetic complexity and scalability of cyclobutyl-modified acceptors.

Based on the above, I recommend a major revision to address these points. My recommendation is for the authors to:

1. Tone down claims about the clamping effect mechanism
2. Address how that morphological improvements are or are not the primary performance driver
3. Add statistical analysis of device performance differences
4. Provide more comprehensive stability data or acknowledge this limitation

The efficiency achievement is noteworthy and the work is technically sound, making it suitable for publication after these revisions address the overclaimed mechanistic insights.

Reviewer #2

(Remarks to the Author)

In this manuscript, the authors reported a small-molecule acceptor substituted with a cyclobutyl group, that can form interchain supramolecular interactions through hydrogen bonding with L8-BO at the external side chains. The authors tried to demonstrate the clamping effect can suppress the electron-phonon coupling and promote the formation of high-quality acceptor alloy phases in ternary active layer with various characterization methods. And optimized carrier behaviors and reduced non-radiative loss were achieved, delivering an efficiency of 20.93%. The results are impressive, however, due to the following issues, e.g., the negligence of the longer butyloctyl groups, the manuscript should be revised significantly before it can be considered for publication.

1. The unit of Stokes shift should be wavenumber (cm⁻¹) instead of nm. So, in page 7, if the authors compare the wavenumbers, I am not sure whether the corresponding deduction can be valid.
2. Besides FWHM, the fluorescence quantum yield in thin film should be another key parameter that is informative for the investigation on electron-phonon coupling.
3. As is well known, the fluorescence quantum yield in solution is normally unaffected by the flexible side chains, why the use of cyclobutyl group can itself reduce the electron-phonon coupling. I mean the cyclobutyl group may only play an important role in suppress the high-frequency vibration of the conjugated backbone by the tight packing induced by the cyclobutyl groups..
4. In Page 12, the slightly larger π - π stacking distance is attributed to the steric hindrance introduced by the rigid cyclobutyl group, which is wrong. Compared with hexyldecyl group on L8-BO, the cyclobutyl group is less sterically hindered. So the possible reason should be the larger butylhexyl group on the nitrogen atoms.
5. The energy level of S-Cb and L8-BO should also be examined by UPS, and be compared with D18.
6. Why did the authors use PFN-Br:MA as the electron transport layer?
7. Since the authors reported a very high PCE, I suggest the value to be examined by the NIM, China.
8. In Page 14, the authors expressed the significant slowdown of charge recombination. what is the basis?
9. Why does the larger LUMO-LUMO energy offset induce the larger ΔE_2 ? The authors should reveal under which conditions the energy losses were evaluated, CT or S-Q theory.
10. The authors should define the δG , what is physical meaning of the term?
11. The clamping effect in Figure 4d and 4e is not indicated clearly, and I suggest the authors to redraw the picture to clearly show the clamping dimer geometry. Especially, the hydrogen bonding should be clearly indicated and confirmed with solid evidence.
12. For the PiFM, why 1426 cm⁻¹ is chosen for S-Cb and 1532 cm⁻¹ for L8-BO. Are they selected for the specific functional groups?
13. The authors demonstrate that ternary system shows better morphological stability under high temperature. I think it is necessary to examine and discuss the stability of OPV devices and make a comparison.
14. Traditionally, the alloy mechanism was characterized by the linear correlation between Voc and the ratio of the third component.

Reviewer #3

(Remarks to the Author)

This study reports a novel cyclobutyl-substituted small-molecule acceptor (S-Cb) that forms interchain supramolecular "clamp" interactions with L8-BO, effectively suppressing electron-phonon coupling and enabling ternary organic solar cells (OSCs) with an impressive efficiency of 20.93% (certified 20.74%). This work is highly innovative, systematic, and thorough. The molecular design is rational, the experimental support is comprehensive, and the combination of morphology characterization, device physics, and theoretical simulations provides a convincing mechanism for the performance enhancement. The reported efficiency is among the highest in the field of OSCs. This manuscript presents a timely and impactful contribution to the field of high-efficiency OSCs. The clamp strategy offers a new pathway to suppress electron-phonon coupling and optimize active layer morphology. The work meets the high standards of Nature Communications and is recommended for acceptance after minor revisions.

1. The introduction of a rigid cyclobutyl side chain to enhance molecular rigidity and suppress electron–phonon coupling is a clever and underexplored strategy. Compare S-Cb with cyclopentyl/cyclohexyl analogues to highlight ring-strain superiority. "Clamping effect" requires clearer definition (e.g., schematic illustrating H-bond positioning).
2. The "clamping effect" between S-Cb and L8-BO is convincingly demonstrated through GIWAXS, PiFM, NMR, and molecular dynamics simulations. Although NMR shows spatial proximity between S-Cb and L8-BO, FTIR or temperature-dependent NMR could further confirm the presence and strength of hydrogen bonding.
3. Quantification of Electron–Phonon Coupling is required. While Stokes shift, FWHM, and Huang–Rhys factors are provided, temperature-dependent PL or charge transport measurements could help quantify the electron–phonon coupling constant more directly.
4. The morphological stability under thermal stress (100°C) is shown in Supplementary Fig. 22, but light-soaking stability should be included to evaluate operational stability.
5. A PCE of 20.93% (certified 20.74%) with low non-radiative loss and good reproducibility represents a state-of-the-art result. The non-radiative loss (ΔE_3) increases again at 75% L8-BO content. Please discuss whether this is related to morphological deterioration or reduced clamp efficiency.
6. Alloy-phase evidence should be consolidated. Specify if "acceptor alloy" implies mixed crystalline/amorphous phases. CV shows linear LUMO shift; DSC shows single crystallisation peak. However, GIWAXS (100) peak drift is $\leq 0.04 \text{ \AA}$, much smaller than high-quality alloys ($\geq 0.1 \text{ \AA}$). Statistical significance (error bars, χ -parameter) is absent. MD $g(r)$ alone is insufficient; orientational order parameter P_2 and Flory–Huggins χ should be included to demonstrate thermodynamic miscibility.
7. Synthetic Scalability: Cyclobutyl synthesis complexity may hinder industrial adoption. A brief cost/feasibility assessment is needed. Industrial viability (e.g., ink formulation, large-area printing compatibility) should be discussed.

Version 1:

Reviewer comments:

Reviewer #1

(Remarks to the Author)

The authors have conducted meaning additional experiments to verify my comments and I can now recommend this manuscript for publication.

Reviewer #2

(Remarks to the Author)

The authors provided extensive revisions to the manuscript and a comprehensive response to my previous comments. The efforts to clarify the scientific content and to elaborate on the material design are evident and have improved the manuscript. However, there remains some concerns need to be addressed.

> There is some conceptual confusion regarding the definitions of radiative energy loss (ΔE_2) and non-radiative energy loss (ΔE_3). Both ΔE_2 and ΔE_3 are linked to the energy offset between the donor and the acceptor. However, in the authors' definitions, ΔE_2 is solely ascribed to this energy offset, whereas ΔE_3 is attributed to other factors such as defects and vibronic coupling. Furthermore, it should be noted that in the current donor-acceptor system, the key factor influencing radiative/non-radiative recombination should be the HOMO offset, rather than LUMO offset. Additionally, Fig. R4 shows that the CT energy level equals the LUMO energy level of the acceptor, and the LE energy level equals the LUMO energy level of the donor. What is the basis for this assumption?

> In DSC measurements, the scanning temperature should typically be maintained below the onset temperature of material degradation. However, in this work, the DSC tests were conducted at temperatures approaching T_d (where the material had already undergone 5% degradation). Notably, for S-Cb, the DSC scanning temperature significantly exceeded T_d .

Reviewer #3

(Remarks to the Author)

The authors have addressed the reviewers' concerns. The revised manuscript is substantially improved and it may be acceptable for publication in its present form.

Reviewers' Comments to Authors:

Reviewer #1:

Wei Gao and colleagues report organic solar cells achieving 20.93% efficiency (20.74% certified) through the incorporation of cyclobutyl-substituted acceptor S-Cb in ternary blends with D18 and L8-BO. They propose that these rigid cyclobutyl groups enable “clamping” interactions with L8-BO via hydrogen bonding, suppressing electron-phonon coupling and improving device performance. This manuscript reports a meaningful efficiency improvement in organic solar cells and employs comprehensive characterization techniques. However, the mechanistic claims, particularly regarding electron-phonon coupling suppression via the “clamping effect” are not definitively proven. The work represents a solid incremental advance through molecular engineering rather than a conceptual breakthrough. Despite this, I recommend publication subject to the following revisions.

Major Comments

1) The proposed mechanism linking cyclobutyl substitution to efficiency improvements involves multiple intermediate effects that are inadequately deconvoluted. The authors show changes in the molecular rigidity (T_g increase of 5.6 °C), crystallinity (4 x higher enthalpy), morphology (altered phase separation), and intermolecular interactions. However, the relative contributions remain unclear and improvements are automatically attributed to the clamping effect of cyclobutyl substitution. The authors need to provide a good statement on these intermediate effects and how much it contributes. For instance in line 198 the authors state 'Both S-Cb and L8-BO films exhibit well-defined face-on orientations, indicating that cyclobutyl substitution does not disrupt the preferential molecular orientation. The emphasis on electron-phonon coupling suppression appears somewhat arbitrary given that morphological improvements alone could explain the performance gains.'

Reply: Thank you for this critical and insightful comment. The increase in T_g upon cyclobutyl substitution primarily indicates the enhanced molecular rigidity of S-Cb

compared to L8-BO. This greater rigidity can suppress vibrations in the alkyl chain region, contributing to the observed smaller Stokes shift and higher PLQY for S-Cb. Correspondingly, energy loss analysis reveals a lower non-radiative energy loss (ΔE_3) in the D18:S-Cb-based device. DSC curves show a higher melting enthalpy for S-Cb than for L8-BO, suggesting a stronger crystallization tendency. Subsequent GIWAXS studies confirm that S-Cb exhibits improved packing order in both neat and D18-blended films. This optimized microstructure leads to higher and more balanced charge carrier mobility in S-Cb neat films and D18:S-Cb blends, which effectively suppresses charge recombination (as verified by TAS). Consequently, the binary devices based on D18:S-Cb achieve a higher J_{sc} (corresponding to a higher EQE response) and FF than their D18:L8-BO counterparts. Through systematic comparison with L8-BO, our comprehensive characterization elucidates the impact of cyclobutyl substitution on S-Cb across multiple aspects: optical absorption, energy levels, suppression of electron-phonon coupling, PLQY, non-radiative energy loss, crystallinity, film morphology, photoelectric conversion processes (including dissociation, transport, and recombination), and ultimately photovoltaic performance.

The hydrogen-bond interaction between the cyclobutyl group in S-Cb and the 2-butyloctyl chain in L8-BO was an unexpected result revealed by molecular dynamics (MD) simulations. In the latter part of the manuscript, we focus on this clamping effect, demonstrating its existence albeit weak strength. We illustrate its role in suppressing electron-phonon coupling and, more importantly, in modulating the blend morphology (including donor-acceptor phase separation and acceptor alloy phase). These effects collectively contribute to the performance enhancement of ternary devices. We apologize for any inconvenience caused by the ordering and logical flow of the discussion in our manuscript.

In the revised manuscript, we have further elaborated the formation mechanism of this hydrogen-bond interaction: the considerable ring strain and resulting bond angle bending in the cyclobutyl group enhance the polarity of the C–H bonds, endowing the hydrogen atoms with a partial positive charge (though still weaker than that of hydrogens directly bonded to oxygen or nitrogen). The compact steric profile and

weakly electrophilic hydrogen of the cyclobutyl group allow it to function as a hydrogen-bond acceptor. Furthermore, we have experimentally confirmed the existence of this interaction using 2D NMR and FTIR. After the “unique dimer” structure was revealed by theoretical calculations, our subsequent work focused on exploring its effects. Considering that this hydrogen bond may facilitate an alternating ordered arrangement of S-Cb \cdots L8-BO \cdots molecules, thereby promoting intermolecular π - π stacking, we first investigated its role in modulating blend morphology. We understand your concern that effective morphology can also be achieved in ternary blends without this clamping effect—which is indeed commonly thought in L8-BO-based ternary systems. However, we note that in typical ternary systems with L8-BO as the main acceptor, the efficiency usually peaks when the third component is around 20 wt%, and further increasing its ratio often leads to performance degradation. In contrast, in D18:S-Cb:L8-BO ternary system, device efficiency improves significantly when the S-Cb ratio reaches 25 wt%, and a further increase to 50 wt% still yields an approximately 0.5% enhancement in efficiency—a phenomenon not observed in other L8-BO-based ternary systems. We tentatively attribute this to the further morphology optimization induced by the clamping effect, which is also supported by GIWAXS and PiFM measurements.

In the revised manuscript, we also quantitatively analyzed the electron-phonon coupling effect by temperature-dependent photoluminescence (PL) spectroscopy of neat S-Cb, neat L8-BO, and their 1:1 blend films. As shown in **Supplementary Fig. 25**, the PL emission peak of the S-Cb:L8-BO (1:1) blend is markedly narrower than those of the neat films, and its electron-phonon coupling constant decreases to 0.163 meV K⁻¹. This significant suppression of electron-phonon coupling stems from the restriction of alkyl chain vibrations in both S-Cb and L8-BO imposed by the clamping effect, while the dimer structure also constrains the overall motion of acceptor molecules. Moreover, thermal stability tests revealed that the performance of D18:S-Cb:L8-BO (1:1) devices first increased and then decreased upon heating. We speculate that this may be due to the gradual disruption of the clamping effect in the initial stage, followed by a faster decay in stability as the ternary morphology evolves toward that

of the D18:L8-BO binary system. This observation indirectly supports the existence and role of the clamping effect. Finally, the clamping effect contributes to significant improvements in PCE of the ternary devices by optimizing blend morphology and effectively suppressing electron-phonon coupling.

Supplementary Fig. 25 Temperature-dependent PL spectra of (a) L8-BO neat film, (b) S-Cb neat film, and (c) S-Cb:L8-BO (1:1) blend film. (d) Linear fitting in the low-temperature region between temperature and FWHM.

2) The hydrogen bonding-mediated clamping between S-Cb and L8-BO feels overstated. MD simulations how the effect only at 50:50 ratio, raising questions about why this specific stoichiometry is special. Should we not see this in higher S-CB concentrations as well? Additionally, the effect of the “unique dimer” has not been adequately explained.

Reply: We thank you for this critical and insightful comment. The stoichiometric specificity of the clamping effect is indeed a central point of our mechanistic proposal. The 50:50 ratio is not arbitrary but is the optimal condition for forming an extended,

cooperative supramolecular network, rather than just isolated dimer pairs. The effect diminishes at other ratios due to molecular competition and packing inefficiencies:

At 50:50 Ratio: The system has the highest probability to form a long-range, alternating structure (S-Cb \cdots L8-BO \cdots S-Cb \cdots L8-BO \cdots). In this configuration, the rigid cyclobutyl "plug" of S-Cb and the branched "clamp" of L8-BO interact complementarily. This allows the hydrogen-bonded clamping to propagate in three dimensions, creating a highly interconnected and stable acceptor alloy phase. Our MD simulations statistically confirm this, showing the population of clamp-type dimers peaks at 27% for the 50:50 blend.

At a high S-Cb Ratio (S-Cb:L8-BO = 75:25): An excess of S-Cb molecules favors homo-dimerization (e.g., the Y-type stacking seen in S-Cb single crystals). Since S-Cb intermolecular interactions are energetically favorable, they outcompete the formation of heteromolecular S-Cb \cdots L8-BO clamp dimers, thereby diluting the clamping network. This is directly observed in our MD data, where the clamp dimer population drops to 16% at a 75:25 ratio.

At a high L8-BO Ratio (S-Cb:L8-BO = 25:75): An excess of L8-BO means there are insufficient rigid S-Cb "plugs" to engage all the flexible L8-BO "clamps." The system forms more disordered, L8-BO-rich mixed regions that lack the structural integrity for effective clamping. Consequently, the clamp dimer population falls drastically to only 8%.

Therefore, the specificity for the 1:1 ratio arises from the optimal complementary and cooperative interaction between the two distinct molecules to form an extended supramolecular structure.

This unique dimer structure, referred to as the clamping effect in our study, is believed to operate through a dual mechanism: it contributes to the suppression of electron-phonon coupling within the system while simultaneously facilitating the optimization of the ternary active layer morphology. Collectively, these effects are responsible for the notable enhancement in PCE observed in the D18:S-Cb:L8-BO-based ternary OSCs devices.

3) The ^1H - ^1H NOESY NMR provides only indirect evidence of proximity, not definitive proof of the proposed binding mode.

Reply: Thank you for this insightful comment. We fully acknowledge that providing direct evidence for the clamping effect poses a significant challenge. Hydrogen bonding is a common intermolecular interaction, which typically requires the presence of both donor and acceptor groups within a suitable spatial proximity. ^1H - ^1H NOESY NMR data indicate that the spatial distance between the relevant hydrogen atoms is within 5 Å, supporting the possibility of an intermolecular interaction. Furthermore, the hydrogen atoms on the cyclobutyl group of S-Cb carry a partial positive charge due to ring strain. Coupled with the relatively small steric hindrance of the cyclobutyl group, this may allow for the formation of a weak C-H...H-C interaction with the BO chain of L8-BO at close range, shown as molecular dynamics simulations. Following the third reviewer's suggestion, we conducted comparative FTIR spectroscopy measurements on neat L8-BO, neat S-Cb, and S-Cb:L8-BO (1:1) blend film. The results show a shift in the C-H stretching vibration peak of the cyclobutyl group in the blend film compared to the neat S-Cb film, providing further experimental indication of a weak hydrogen-bond interaction. The combination of 2D NMR and IR spectroscopy to indicate such weak interactions is a commonly accepted approach in the literature (*J. Am. Chem. Soc.* 2022, 144, 14731). While we acknowledge that definitive confirmation may require further investigation, the consistency across multiple experimental datasets and analyses supports its existence.

Thanks again for your valuable comments on this interesting phenomenon. We will continue to explore this aspect and hope to gain more direct insights in future work.

4) I find that the correlation between computed Huang-Rhys factors and device performance is tenuous.

Reply: We thank you for this critical comment, which allows us to further clarify the fundamental physical principle that forms the cornerstone of our mechanistic interpretation. The connection between the Huang-Rhys factor and device performance

is actually not tenuous (although very less studied); rather, it is directly mandated by Marcus charge transfer theory, and our experimental device data provide a compelling validation of this theoretical prediction.

The core of our argument rests on the well-established relationship between electron-phonon coupling and non-radiative recombination in organic semiconductors. According to Marcus theory (as applied to non-radiative decay from the charge-transfer state to the ground state), the non-radiative recombination rate (k_{nr}) is exponentially dependent on the reorganisation energy (λ). The critical link is that the total reorganisation energy is the sum of the contributions from all vibrational modes, $\lambda = \sum S_i \hbar \omega_i$, where S_i are the Huang-Rhys factors and $\hbar \omega_i$ are the vibrational energies. A lower Huang-Rhys factor (S) directly translates to a smaller reorganisation energy (λ). This relationship has profound consequences for device performance, as it directly governs the non-radiative voltage loss (ΔV_{nr}):

$$k_{nr} = \frac{2\pi}{\hbar} V_{CT,s_0}^2 FCWD(0)$$

where $FCWD(\hbar\omega)$

$$= \frac{1}{\sqrt{4\pi\lambda_0 k_B T}} \sum_{w=0}^{\infty} \sum_{t=0}^{\infty} \frac{e^{-S} S^{w-t} t!}{w!} [L_t^{w-t}(S)]^2 e^{-\{[\hbar\omega - \Delta G_0 + \lambda_0 + (w-t)\hbar\Omega]^2 / 4\lambda_0 k_B T\}} \frac{1}{e^{t\hbar\Omega/k_B T} + 1}$$

$$\Delta V_{nr} = -\frac{k_B T}{q} \ln(EQE_{EL})$$

The electroluminescence quantum efficiency (EQE_{EL}) itself is proportional to the ratio of radiative to total (radiative + non-radiative) recombination rates. A smaller λ (resulting from lower S) leads to a drastic reduction in k_{nr} . This is not a matter of speculation but a foundational principle in the field (see, for example, our reference Chen, X.-K. et al. *Nat. Energy* **6**, 799-806 (2019), Ref. 53 in our manuscript).

Our work provides a solid validation of this theory from computation to device metrics: Our calculations reveal that the "clamp-type" dimer exhibits significantly lower average and maximum Huang-Rhys factors (Fig. 4c, Supplementary Fig. 24) compared to all other π - π stacking motifs. This computationally predicts a suppressed intrinsic k_{nr} . We directly measured the EQE_{EL} of our devices (Fig. 2f). The results

confirm the theoretical prediction: the optimal 50:50 ternary blend, where the clamp-dimer population is maximized, shows a dramatic enhancement in EQE_{EL} (from $\sim 1.9 \times 10^{-4}$ to $\sim 3.1 \times 10^{-4}$). This enhanced EQE_{EL} translates directly into a lower non-radiative energy loss (ΔE_3) of 0.210 eV for the champion device, compared to 0.222-0.230 eV for the binary devices (Supplementary Table 3). The suppression of this loss channel is a primary contributor to the high V_{OC} achieved concurrently with improvements in J_{SC} and FF, leading to the record PCE.

5) While 10-15 devices were tested per condition, no statistical analyses determine whether performance differences are significant given the reported standard deviations (0.16-0.27%). The optimal 50:50 ratio could be within experimental error of other compositions

Reply: Thank you for this critical comment. We fully agree that the rigor of data statistics is essential for supporting research conclusions. In one experiment, we fabricated the same number of devices for each condition (with different L8-BO ratios). However, since not all devices functioned properly, our statistical analysis was based only on data from operational devices, which leads to variations in the sample sizes for ternary devices based on D18:S-Cb:L8-BO with different L8-BO contents. **Supplementary Fig. 13** shows the statistical distribution of key parameters from the operational devices within the same batch, and **Table 1** presents the average PCE along with the mean squared error for each condition. Significance analysis was performed based on the number of measurements, mean values, and standard deviations for each dataset. F-tests and t-tests conducted between groups showed no statistically significant differences, indicating the absence of systematic error between different data groups and the reliability of the data. Even though systematic error exists, since all experimental data were collected by the same operator under identical testing conditions, potential systematic errors remained consistent across groups. We believe that calculating the average values already helps to reduce the influence of random errors to a considerable extent, thereby allowing us to observe performance trends

across devices with different L8-BO ratios. The mean squared error reflects the dispersion of measured values around the average, which can be used to evaluate the repeatability and consistency of device fabrication. According to the statistical results, devices based on the D18:S-Cb:L8-BO system showed the best repeatability. The repeatability of other systems was slightly lower but still within an acceptable range, overall following general statistical expectations. Furthermore, based on the mean values, the ternary device with 50% L8-BO shows an approximately 0.6% higher efficiency compared to those with 25% and 75% L8-BO.

We have revised it in the manuscript accordingly:

Based on device data of each sample condition (10 to 15 devices each), the mean, and standard deviation, F- and t-test statistical analysis of the photovoltaic parameters across different L8-BO content indicated no statistically significant differences between the groups.

6) The study examines only one system. Claims about the broader applicability of cyclobutyl-mediated interactions require demonstration in other donor-acceptor combinations. Should we expect the same behaviour with other binary and ternary blend as well? Or is the clamping effect only applicable to this particular combination?

Reply: Thank you for the valuable suggestion. We have replaced the L8-BO with BTP-eC9 and fabricated the ternary device based on D18:S-Cb:BTP-eC9 with a BTP-eC9 content of 50%. The corresponding *J-V* curves and EQE spectra are provided in **Supplementary Fig.14**, and the parameters are summarized in **Supplementary Table 2**. The device data revealed that the ternary device showed improvements in both J_{sc} and FF, with a particularly notable enhancement in FF compared to the D18:BTP-eC9 binary device. Consequently, the power conversion efficiency (PCE) increased from 19.21% to 20.23%. The extent of this improvement is comparable to that previously observed when transitioning from the D18:L8-BO device to the D18:S-Cb:L8-BO device. Furthermore, to investigate the intermolecular interactions, we performed ^1H - ^1H NOESY NMR measurements on S-Cb:BTP-eC9 (1:1) blend solution

(**Supplementary Fig. 26e**). The spectrum revealed correlation peaks indicating interactions between the cyclobutyl side chain of S-Cb and straight chain of BTP-eC9. This result provides further evidence for the proposed clamping effect and suggests that this effect can effectively for systems featuring both branched and linear alkyl chains.

We have added the data and discussion in the manuscript accordingly:

To further verify the general applicability of S-Cb as a ternary component, we replaced L8-BO with another high-performance acceptor, BTP-eC9, to fabricate ternary devices based on D18:S-Cb:BTP-eC9 with a BTP-eC9 content of 50 wt%. The $J-V$ curves and EQE spectra of the champion device are presented in **Supplementary Fig. 14**, with the corresponding photovoltaic parameters summarized in **Supplementary Table 2**. The binary device based on D18:BTP-eC9 yielded a PCE of 19.21%, slightly lower than that of D18:S-Cb-based OSCs. In contrast, the ternary device based on D18:S-Cb:BTP-eC9 (1:1) achieved an enhanced PCE of 20.23%, with an V_{oc} of 0.876 V, a J_{sc} of 28.54 mA cm^{-2} , and an FF of 80.9%. The efficiency enhancement, primarily attributed to the notable improvement in FF, represents an absolute increase of approximately 1%. The magnitude of this gain is comparable to that observed when introducing S-Cb into the D18:L8-BO binary system. These results demonstrate that cyclobutyl-substituted S-Cb serves as a versatile ternary component, exhibiting excellent compatibility with acceptor materials bearing either linear or branched alkyl side chains and effectively boosting device performance.

To verify the generality of the clamping effect induced by the cyclobutyl group, specifically probing whether effective long-range interactions exist with the linear alkyl chains on acceptor molecule, we performed $^1\text{H}-^1\text{H}$ NOESY NMR spectroscopy on a 1:1 blend of S-Cb and BTP-eC9. The relevant hydrogen atoms on BTP-eC9 are labeled as Hi-Hk, as illustrated in **Supplementary Fig. 26a**. The NOESY spectrum reveals distinct spatial proximity correlations between the hydrogen atoms of the cyclobutyl group on S-Cb (Ha) and those on the linear alkyl chains of BTP-eC9 (Hj and Hi) (**Supplementary Fig. 26e**). The intensity of this interaction appears stronger than that previously observed in the S-Cb:L8-BO system, which can be attributed to the lower

steric hindrance of linear alkyl chains compared to branched ones that allows the cyclobutyl group to approach the main body of the linear chain more closely. This result provides evidence for a favorable spatial proximity and orientation between the cyclobutyl group of S-Cb and the linear alkyl chain-based acceptor, forming a crucial structural foundation for the clamping effect. This finding further reinforces that the cyclobutyl-induced clamping effect of S-Cb may be a general phenomenon applicable to acceptor molecules functionalized with either branched or linear alkyl side chains.

Supplementary Fig. 14 (a) Optimal $J-V$ curves for OSCs based on D18:BTP-eC9 and D18:S-Cb:BTP-eC9 (1:1). (b) Corresponding EQE spectrum for OSCs based on D18:S-Cb:BTP-eC9 (1:1).

Supplementary Fig. 26 (a) Hydrogen atoms labeling Ha-Hk and their positions on S-Cb, L8-BO, and BTP-eC9. ^1H - ^1H NOESY NMR spectra of S-Cb:BTP-eC9 (0.5:0.5).

Minor Comments

1. The single crystal analysis, while thorough, may have limited relevance to solution-processed films with substantial disorder.

Reply: Thank you for the insightful comments. The relationship between the single-crystal structure and the thin-film morphology is indeed a critical issue, as you rightly pointed out. We fully appreciate your concern: the growth of single crystals typically requires an extended period to achieve thermodynamically equilibrated, ordered packing, whereas the fabrication of active layers (e.g., via spin-coating) is a rapid kinetic process. These significant differences in formation conditions make it essential to examine whether and to what extent the molecular packing observed in single crystals is preserved in thin films.

It is worth noting that relevant studies offer some perspective on this matter. For instance, in the case of the classic non-fullerene acceptor Y6, the primary π - π stacking features observed in its single crystals have been identified in corresponding thin films using GIWAXS (*Adv. Energy Mater.* 2022, 12, 2201338). This indicates that certain core molecular packing motifs can be transferred from the single-crystal state to the film state. A salient example is the significant difference in molecular packing between L8-BO and Y6. L8-BO exhibits bimolecular packing configurations and a tighter packing framework distinctly different from those of Y6. This inherent propensity for tighter packing, revealed by single-crystal analysis, is considered one of the important reasons why devices based on L8-BO achieve higher FF and photovoltaic performance. Therefore, we would like to state that while due to the fundamental differences in their formation processes, this correlation may not be direct or strongly predictive and need to be kept in mind, the single-crystal structures can indeed provide valuable insights into molecular arrangement in films..

2. PiFM characterization is sophisticated but interpretation of "acceptor alloy phase" from IR absorption alone is problematic.

Reply: Thank you for the insightful comment. In elucidating the working mechanism

of the ternary device, we have provided multi-faceted evidence supporting the formation of an acceptor alloy phase between S-Cb and L8-BO, based mainly on the following points: 1) considering the intrinsic material properties, S-Cb and L8-BO share similar molecular structures, and their calculated surface energies are very close. According to the principle of “like dissolves like,” this provides a thermodynamic foundation for the formation of a homogeneous alloy phase upon blending. 2) device performance and energy level measurements offer direct evidence. As the content of the high- V_{oc} component, L8-BO, increases in the ternary system, the device’s V_{oc} shows a linear increase. Concurrently, the HOMO and LUMO levels of the S-Cb:L8-BO blend film also shift linearly with the L8-BO content. Both linear relationships are characteristic features of alloy phase formation. 3) thermal analysis provides further support. DSC measurements indicate that the mixture of S-Cb and L8-BO exhibits a single melting peak at a temperature lower than that of either pure component. This strongly suggests the formation of a new, homogeneous alloy phase in the solid state. The results above point to the formation of an acceptor alloy phase between S-Cb and L8-BO.

The strength of PiFM lies in its ability to perform selective imaging of individual components based on specific infrared wavelengths. We obtained pure-component morphology images for S-Cb and L8-BO within D18:S-Cb:L8-BO blend films at different L8-BO contents (25%, 50%, 75%), corresponding to Fig. 5 g,h; 5 l,m; and 5 q,r, respectively. Analysis revealed that at a high L8-BO content (75%), the pure-component morphology images for both materials show relatively distinct, discontinuous pore-like structures (Fig. 5 q,r). When the L8-BO content is reduced to 25%, these pore structures become less pronounced (Fig. 5 g,h). Notably, at the same ratio of 50%, no obvious pore structures are observed in the morphology images of either S-Cb or L8-BO, which instead present more uniform and continuous features (Fig. 5 l,m). We interpret this phenomenon as follows: if S-Cb and L8-BO have poor compatibility and fail to form a homogeneous alloy, they tend to self-aggregate separately. This would expose more phase-separated boundaries and pore structures in

their respective pure-component images. Conversely, when the two components mix well and form a uniform alloy phase, molecular-level intermixing suppresses the separate aggregation of each pure component. Consequently, when visualizing the distribution of S-Cb or L8-BO individually, the morphology appears continuous and homogeneous. Therefore, by leveraging PiFM's single-component imaging capability and analyzing the uniformity and continuity of the respective morphologies of S-Cb and L8-BO, we can infer their degree of intermixing at the nanoscale. This provides morphological evidence supporting the formation of an acceptor alloy phase.

3. Stability testing (100°C, 1 day) is insufficient for practical relevance. Light stability and longer-term studies are needed for a proper conclusion.

Reply: Thank you for the valuable suggestion. Following your advice, we have supplemented the manuscript with an investigation into the photothermal stability of the D18:S-Cb:L8-BO system. We fabricated and tested inverted ternary devices based on D18:S-Cb:L8-BO with L8-BO contents of 0%, 50%, and 100% to evaluate their performance degradation under continuous light soaking and heating. The corresponding stability data have been added to the revised manuscript in **Supplementary Fig. 33**. This data provides insights for understanding the long-term operational reliability of this system. Concurrently, we have also objectively realized the current limitations in device stability within the manuscript.

We have added the data and discussion in the manuscript accordingly:

To further investigate the impact of the clamping effect on thermal- and photo-stability of OSCs, we fabricated inverted devices with a structure of ITO/ZnO/active layer/MoO_x/Ag. The stability test results for devices based on D18:S-Cb:L8-BO with 0%, 50%, and 100% L8-BO content are presented in **Supplementary Fig. 33**. Under continuous annealing at 100 °C, D18:S-Cb device exhibited significantly better thermal stability than the D18:L8-BO device, which can be attributed to the higher T_g of S-Cb resulting from its short and rigid cyclobutyl substituent. Furthermore, when 50% L8-BO was incorporated into the D18:S-Cb active layer, the device stability showed

improvement within the first 160 h of heating. However, beyond 160 h, the thermal stability of the ternary device declined more rapidly compared to the D18:S-Cb binary device. We speculate that during the early stage of heating, the clamp effect mediated by hydrogen bonding helps to stabilize the morphology. With prolonged thermal stress, this hydrogen-bond network may gradually degrade, causing the ternary blend to evolve toward a morphology resembling that of D18:L8-BO, thus accelerating the decay in device performance.

It is worth noting that, under continuous heating at 100 °C, these weak interactions provide limited stabilization, which explains why the stability improvement is modest and eventually deteriorates over extended periods, leading to the observed trend of initial stability followed by rapid decay in the ternary system. In contrast, under continuous illumination, such hydrogen-bonding interactions appear less susceptible to degradation. Consequently, the device based on D18:S-Cb:L8-BO with 50% L8-BO content demonstrated markedly better photo-stability than the D18:S-Cb binary device. Although the overall thermal- and photo- stability of the ternary devices remains inferior to state-of-the-art values reported in the literature⁵⁶—particularly in terms of thermal stability—these results provide indirect evidence supporting the role of the hydrogen-bond-guided clamping effect in modulating blend morphology and device stability.

Supplementary Fig. 33 (a) Thermal- and (b) photo-stability test of OSCs based on D18:L8-BO, D18:S-Cb and D18:S-CbLL8-BO (0.5:0.5).

4. The Stokes shift reduction (2.3 nm) is marginal and within measurement uncertainty.

Reply: We thank you for the comment. We highly value the point raised regarding the potential uncertainty in our absorption and photoluminescence (PL) spectral measurements. The spectral measurements were conducted with an instrumental step size of 0.5 nm. We acknowledge that this setting limitation may impose certain limitations on precisely determining minute peak shifts. Nevertheless, under the current measurement conditions, it remains reliable for identifying the main peak positions of the absorption and emission spectra, which provides a baseline for calculating the Stokes shift. To further enhance the precision of our analysis and in direct response to the valuable suggestion from the second reviewer, we have uniformly converted the unit of the Stokes shift to wavenumbers in the revised manuscript. This approach intrinsically mitigates potential errors associated with the linear wavelength scale.

Furthermore, beyond the Stokes shift, we analyzed other key spectral parameters. The data reveal a significant difference in the full width at half maximum (FWHM) of the PL spectra between pristine S-Cb and L8-BO films. For a more quantitative understanding, we also performed temperature-dependent PL measurements on pristine S-Cb and L8-BO films (**Supplementary Fig. 25**). This analysis quantitatively elucidates a clear difference in their electron-phonon coupling strengths, and the results are fully consistent with the trend inferred from the Stokes shift analysis. These multi-faceted spectroscopic evidences corroborate each other and collectively support our conclusions.

5. Fig. 4f showing free volume is difficult to interpret. Quantitative analysis would be preferable to color maps. I can't really see where the free volume is.

Reply: Thank you for the valuable suggestion. We have added corresponding color annotations to Fig. 4 in the manuscript to improve readability. The annotations are specified as follows: the cyan region represents the free volume; the dark blue region corresponds to the donor material D18; the orange region represents S-Cb, and the yellow region corresponds to L8-BO.

Fig. 4 | Molecular dynamics simulations. (a) Dimer pair distance (r) as a function of pair distribution $g_{S-Cb-L8-BO}(r)$; (b) Dimer types pair counts in different L8-BO ratios in D18:S-Cb:L8-BO ternary blend. (c) Huang Rhys factor of dimer types. (d) Weak interaction in clamp-type dimer. (e) Coulomb attractive energy and electron-hole separation distance in clamp-type dimer. (f) Simulated free volume and annealing donors and acceptors blend at 150°C.

6. The manuscript would benefit from discussing synthetic complexity and scalability of cyclobutyl-modified acceptors.

Reply: Thank you for the valuable suggestion. To discuss the synthetic complexity and scalability of the cyclobutyl-substituted acceptor (S-Cb), we have conducted a comparative analysis of its estimated cost against that of L8-BO, which is widely regarded as one of the state-of-the-art electron acceptors currently available, in direct response to your comment. The key to scaling up the production of L8-BO lies in the high-yield synthesis of the 3-(2-butyloctyl)thieno[3,2-*b*]thiophene monomer. Generally, there are three common synthetic routes, as shown below:

Fig. R1. Synthetic route 1 for 3-(2-butyloctyl)thieno[3,2-*b*]thiophene.

Fig. R2. Synthetic route 2 for 3-(2-butyloctyl)thieno[3,2-*b*]thiophene.

Fig. R3. Synthetic route **3** for 3-(2-butyloctyl)thieno[3,2-*b*]thiophene.

In synthetic route **1**, reagents such as PCC (containing heavy metal Cr³⁺), diethyl ether (highly explosive), and lithium aluminum hydride (flammable) are required, and the process involves up to 7 steps from the starting material to the target compound. Moreover, the yield of the dehydroxylation step (below) in this route is difficult to control consistently.

Therefore, route **1** may face significant challenges for future large-scale production.

In synthetic route **2**, when the reaction scale exceeds 10 grams, the yield of the step (below) decreases considerably.

However, by employing the reaction conditions corresponding to synthetic route **3**, the yield of this step (below) can exceed 90%.

We have successfully scaled up the synthesis to **50 grams** with maintained yield. Based on our experimental experience, route **3** (currently used for synthesizing S-Cb) appears to be the most feasible approach for the industrial-scale synthesis of 3-(2-butyloctyl)thieno[3,2-*b*]thiophene in the future.

When preparing L8-BO and S-Cb via route **3**, the only difference in starting materials is that L8-BO requires 2-butyloctyl bromide (Br-BO), while S-Cb requires (bromomethyl)cyclobutene (Br-Cb). Currently, the market price of Br-BO (CAS No.: 85531-02-8) is about \$3.2 per gram, whereas Br-Cb (CAS No.: 17247-58-4) costs approximately \$1.1 per gram (the alkyl chain of Br-Cb is commercially available in

large quantities). Additionally, considering that S-Cb uses a BO chain as the alkyl group linked to the nitrogen atom in the subsequent synthesis route, while L8-BO employs a 2-ethylhexyl (EH) chain, the overall cost of the alkyl chain combinations in both materials is comparable. Given that the current market price of L8-BO is around \$450 per gram, we estimate that S-Cb would be priced at a similar level.

We have added the discussion into the revised manuscript accordingly:

Evaluation reveals that if L8-BO is synthesized via the same route as S-Cb, the alkyl chain combinations in both materials would lead to comparable large-scale production costs for S-Cb relative to L8-BO. It is worth noting that (bromomethyl)cyclobutene (the carboxylic acid precursor containing one extra carbon atom) required for S-Cb is commercially available at a lower price than the 2-butyloctyl bromide used for L8-BO.

Based on the above, I recommend a major revision to address these points. My recommendation is for the authors to:

1. Tone down claims about the clamping effect mechanism
2. Address how that morphological improvements are or are not the primary performance driver.
3. Add statistical analysis of device performance differences.
4. Provide more comprehensive stability data or acknowledge this limitation

The efficiency achievement is noteworthy and the work is technically sound, making it suitable for publication after these revisions address the overclaimed mechanistic insights.

Reply: We sincerely appreciate your recognition of our work and the time you have invested in reviewing our manuscript. Thank you for your valuable and constructive comments. We have revised the manuscript carefully and thoroughly based on your suggestions and the feedback from the other reviewers and enhanced overall quality.

Reviewer #2:

In this manuscript, the authors reported a small-molecule acceptor substituted with a cyclobutyl group, that can form interchain supramolecular interactions through hydrogen bonding with L8-BO at the external side chains. The authors tried to demonstrate the clamping effect can suppress the electron-phonon coupling and promote the formation of high-quality acceptor alloy phases in ternary active layer with various characterization methods. And optimized carrier behaviors and reduced non-radiative loss were achieved, delivering an efficiency of 20.93%. The results are impressive, however, due to the following issues, e.g., the negligence of the longer butyloctyl groups, the manuscript should be revised significantly before it can be considered for publication.

1. The unit of Stokes shift should be wavenumber (cm⁻¹) instead of nm. So, in page 7, if the authors compare the wavenumbers, I am not sure whether the corresponding deduction can be valid.

Reply: Thank you for the valuable suggestion. As advised, we have recalculated the Stokes shifts for L8-BO and S-Cb neat films using the energy-based formula in wavenumbers: Stokes shift = $(1/\lambda_{\text{abs}} - 1/\lambda_{\text{em}}) \times 10^7 \text{ cm}^{-1}$. The results show Stokes shift values of 1420 cm⁻¹ for L8-BO and 1382 cm⁻¹ for S-Cb. This outcome confirms the trend observed initially with nanometer units, namely that the shift for S-Cb is slightly smaller than that for L8-BO. It is worth noting that by employing the wavenumber unit, the difference between the two values is presented more distinctly.

We have updated the data in the manuscript accordingly:

By combining these values with absorption peaks (**Supplementary Table 1**), the Stokes shifts of L8-BO and S-Cb neat films were calculated to be 1420 cm⁻¹ and 1382 cm⁻¹, respectively.

2. Besides FWHM, the fluorescence quantum yield in thin film should be another key parameter that is informative for the investigation on electron-phonon coupling.

Reply: Thank you for the valuable suggestion. We conducted six parallel measurements of the PLQY for both S-Cb and L8-BO neat films to ensure data reliability with the results presented in **Fig. 1d**.

We have added the data and discussion in the manuscript accordingly:

Parallel measurements of the photoluminescence quantum yield (PLQY) for L8-BO and S-Cb neat films were conducted, with the results from six independent measurements shown in **Fig. 1d**. The average PLQY was 9.9% for L8-BO and 10.7% for S-Cb. The slightly higher PLQY of S-Cb suggests that the cyclobutyl substituent helps to suppress electron-phonon coupling to reduce the non-radiative transition rate (k_{nr}) to some extent, according to the relationship $PLQY = k_r / (k_r + k_{nr})$, where k_r is the radiative transition rate.

Fig. 1 | Chemical structures and photophysical properties. (a) Chemical structures of D18, S-Cb, and L8-BO. (b) Deviation metric from temperature-dependent UV-vis absorption spectra of S-Cb and L8-BO neat films. (c) Normalized absorption and PL spectra of S-Cb and L8-BO neat films. (d) DSC curves of S-Cb and L8-BO. (e) 2D GIWAXS patterns of S-Cb and L8-BO neat films. (f) Corresponding 1D line-cut

profiles for S-Cb and L8-BO neat films. (g) CV curves of S-Cb and L8-BO films.

3. As is well known, the fluorescence quantum yield in solution is normally unaffected by the flexible side chains, why the use of cyclobutyl group can itself reduce the electron-phonon coupling. I mean the cyclobutyl group may only play an important role in suppress the high-frequency vibration of the conjugated backbone by the tight packing induced by the cyclobutyl groups.

Reply: We sincerely thank the reviewer for the insightful comments, which prompted us to further investigate the precise mechanism by which the cyclobutyl substituent suppresses electron-phonon coupling: whether it primarily stems from the suppression of molecular vibrations due to its own rigidity or results from molecular packing.

To clarify this, we measured the steady-state PL spectra and PLQY of S-Cb and L8-BO in dilute solution (**Supplementary Fig. 4**). At the same concentration (3.0×10^{-6} mmol mL⁻¹), the PL intensity of S-Cb is 2.4 times that of L8-BO. Comparison of normalized absorption and PL spectra reveals that S-Cb has a smaller Stokes shift (838 cm⁻¹) than L8-BO (862 cm⁻¹) and a narrower emission FWHM (55.9 nm for S-Cb vs. 61.0 nm for L8-BO). Furthermore, the PLQY of S-Cb in dilute solution reaches 29.16%, approximately 1.8 times that of L8-BO (16.61%). Considering the negligible molecular aggregation in dilute solution, these results strongly suggest that the inherent rigidity of the cyclobutyl group can effectively suppress intramolecular electron-phonon coupling, thereby reducing exciton-molecular vibration interaction and mitigating non-radiative transition rates caused by vibrational relaxation.

We infer that the suppressive effect of cyclobutyl on molecular vibrations also exists in the solid-state film. However, the transition from solution to film state leads to a more pronounced decrease in PLQY for S-Cb compared to L8-BO, indicating that S-Cb may undergo a more significant aggregation-caused quenching (ACQ) effect. This could be attributed to the cyclobutyl substitution promoting more pronounced *H*-aggregation as suggested by the more distinct *0-1* vibrational shoulder in its absorption spectrum, which enhances electron-phonon coupling and accelerates energy dissipation.

The weaker overall electron-phonon coupling observed in S-Cb compared to L8-BO is likely primarily due to the effective reduction of molecular vibrations by the rigid cyclobutyl group. Simultaneously, its shorter alkyl chain might also enhance molecular packing (particularly *H*-aggregation), which exacerbates ACQ and partially counteracts the benefits gained from rigidity. It is worth noting that in the blend film (D18:S-Cb), theoretical calculations indicate a smaller free volume and a lower Huang-Rhys factor compared to D18:L8-BO, providing computational support for the suppression of electron-phonon coupling.

We have added the data and discussion in the manuscript accordingly:

To decouple whether this suppression originates from the inherent molecular rigidity or the molecular packing effects, we further examined their properties in dilute solution (**Supplementary Fig. 4**). At the same concentration (3.0×10^{-6} mmol mL⁻¹), the PL intensity of S-Cb is 2.4 times that of L8-BO. Comparison of normalized absorption and PL spectra reveals that S-Cb has a smaller Stokes shift (838 cm⁻¹ for S-Cb vs. 862 cm⁻¹ for L8-BO) and a narrower emission FWHM (55.9 nm for S-Cb vs. 61.0 nm for L8-BO). Most notably, the PLQY of S-Cb in dilute solution reaches 29.2%, nearly double that of L8-BO (16.6%). Considering the negligible molecular aggregation in dilute solution, these results strongly suggest that the inherent rigidity of the cyclobutyl group can effectively suppress intramolecular electron-phonon coupling, thereby reducing exciton-molecular vibration interaction and mitigating k_{nr} caused by vibrational relaxation.

We posit that this intramolecular vibrational suppression effect of the cyclobutyl group persists in the solid state. However, the transition from solution to film leads to a much more pronounced decrease in PLQY for S-Cb than for L8-BO, revealing that S-Cb undergoes a more significant aggregation-caused quenching (ACQ) effect in the solid state, likely attributable to the cyclobutyl substitution promoting a stronger propensity for *H*-aggregation as corroborated by the more distinct *0-1* vibrational shoulder in its absorption spectrum (**Fig. 1c**), which will exacerbate electron-phonon coupling and facilitate non-radiative energy dissipation⁴⁷. The weaker overall electron-

phonon coupling observed in S-Cb compared to L8-BO is likely primarily due to the effective reduction of molecular vibrations by the rigid cyclobutyl group. Simultaneously, its shorter alkyl chain might also enhance molecular packing (particularly *H*-aggregation), which exacerbates ACQ and partially counteracts the benefits gained from rigidity.

Supplementary Fig. 4 (a) PL spectra of L8-BO and S-Cb in dilute solution with the same concentration. (b) Normalized absorption and PL spectra of L8-BO and S-Cb in solution. (c) PLQY results of L8-BO and S-Cb in solution.

4. In Page 12, the slightly larger π - π stacking attributed to the steric hindrance introduced by the rigid cyclobutyl group, which is wrong. Compared with hexyldecyl group on L8-BO, the cyclobutyl group is less sterically hindered. So the possible reason should be the larger butylhexyl group on the nitrogen atoms.

Reply: Thank you for pointing out the inaccurate description in our manuscript. We believe that the stronger π - π stacking observed in S-Cb is attributed to the 2-butyloctyl

group attached to the N atom. We have revised the manuscript accordingly to correct the previous inappropriate statement and appreciate your valuable feedback.

We have corrected the relevant statement in the manuscript accordingly:

Analysis of the 1D profiles reveals that π - π stacking distance of S-Cb (3.62 Å) is slightly larger than that of L8-BO (3.57 Å), likely due to the larger steric hindrance introduced by the 2-butyloctyl group attached to the N atom of S-Cb.

The slightly lower electron mobility of S-Cb is consistent with the GIWAXS results, as the steric hindrance from the 2-butyloctyl group in S-Cb increases the π - π stacking distance, thereby impeding charge transport.

5. The energy level of S-Cb and L8-BO should also be examined by UPS, and be compared with D18.

Reply: We thank you for the valuable suggestion. In response to the concern regarding the accurate determination of energy levels, we have performed additional UPS measurements on the neat films of D18, L8-BO, and S-Cb with results provided in **Supplementary Fig. 10**. For the estimation of the LUMO energy levels, we employed the formula $E_{\text{LUMO}} = E_{\text{HOMO}}^{\text{UPS}} + E_{\text{g}}^{\text{fund}}$, where the fundamental gap ($E_{\text{g}}^{\text{fund}}$) could be determined from cyclic voltammetry (CV) measurements, i.e., $E_{\text{g}}^{\text{fund}} = E_{\text{LUMO}}^{\text{CV}} - E_{\text{HOMO}}^{\text{CV}}$. This quantity encompasses both the optical gap ($E_{\text{g}}^{\text{opt}}$) and the exciton binding energy (E_{b}), i.e., $E_{\text{g}}^{\text{fund}} = E_{\text{g}}^{\text{opt}} + E_{\text{b}}$. Our rationale for using $E_{\text{g}}^{\text{fund}}$ is as follows: the physical meaning of $E_{\text{g}}^{\text{fund}}$ corresponds to the minimum energy required to completely remove an electron from the HOMO level out of the system, which is the difference between the ionization energy (IE) and the electron affinity (EA). Consequently, we utilized the $E_{\text{g}}^{\text{fund}}$, rather than the $E_{\text{g}}^{\text{opt}}$, provides a more physically rigorous basis for estimating the LUMO level associated with free charge carriers.

We have added the data and discussion in the manuscript accordingly:

The highest occupied molecular orbital (HOMO) energy levels of D18, L8-BO, and S-Cb neat films were determined by ultraviolet photoelectron spectroscopy (UPS), as shown in **Supplementary Fig. 10**. The measured HOMO levels are -5.09 eV, -5.20 eV,

and -5.14 eV for D18, L8-BO, and S-Cb, respectively. To accurately calculate the corresponding lowest unoccupied molecular orbital (LUMO) levels, cyclic voltammetry (CV) measurements were further performed on L8-BO and S-Cb neat films (**Supplementary Fig. 11**). The HOMO^{CV}/LUMO^{CV} levels derived from CV curves are -5.63/-3.88 eV for L8-BO and -5.57/-3.93 eV for S-Cb. The fundamental gap ($E_g^{\text{fund}} = E_{\text{LUMO}^{\text{CV}}} - E_{\text{HOMO}^{\text{CV}}}$) is thus determined to be 1.75 eV for L8-BO and 1.64 eV for S-Cb, while that of D18 (2.24 eV) is taken from our previous report⁴⁰. Consequently, the LUMO levels were calculated using $E_{\text{LUMO}} = E_{\text{HOMO}}^{\text{UPS}} + E_g^{\text{fund}}$, yielding values of -2.85 eV for D18, -3.45 eV for L8-BO, and -3.50 eV for S-Cb. It is noteworthy that the E_g^{fund} physically represents the minimum energy required to completely remove an electron from the HOMO level, corresponding to the difference between the ionization energy (IE) and the electron affinity (EA). Therefore, using E_g^{fund} instead of the optical gap (E_g^{opt}) provides a more rigorous basis for estimating the LUMO level. The trend in HOMO and LUMO levels revealed by UPS and CV is consistent, providing mutual verification. Moreover, the energy levels of L8-BO and S-Cb align well with those of D18 (**Fig. 1g**), which is expected to create an effective driving force for charge separation at the donor/acceptor interface. However, the slightly lower LUMO level of S-Cb compared to that of L8-BO may adversely affect the enhancement of the V_{oc} .

Supplementary Fig. 10 UPS measurements for (a) D18, (b) L8-BO, and (c) S-Cb.

Fig. 1 | Chemical structures and photophysical properties. (a) Chemical structures of D18, S-Cb, and L8-BO. (b) Deviation metric from temperature-dependent UV-vis

absorption spectra of S-Cb and L8-BO neat films. (c) Normalized absorption and PL spectra of S-Cb and L8-BO neat films. (d) DSC curves of S-Cb and L8-BO. (e) 2D GIWAXS patterns of S-Cb and L8-BO neat films. (f) Corresponding 1D line-cut profiles for S-Cb and L8-BO neat films. (g) CV curves of S-Cb and L8-BO films.

6. Why did the authors use PFN-Br:MA as the electron transport layer?

Reply: We thank you for raising this insightful point. The rationale for employing melamine (MA) as a dopant is based on the following considerations: MA is an organic compound featuring a high nitrogen content and exhibits good solubility in alcohols such as methanol and ethanol. This property ensures the formation of a homogeneous blended solution with PFN-Br. More importantly, the amino groups (-NH₂) within the MA molecular structure can form hydrogen bonds with functional groups in the organic semiconductor materials of the active layer. This intermolecular interaction facilitates the establishment of more effective contact at the interface between the active layer and the cathode interfacial layer. As supported by the literature (*ACS Energy Lett.* **2021**, 6, 3582-3589), incorporating MA into PFN-Br has been demonstrated to modulate the interface dipole between the active layer and the cathode layer, as well as to optimize the Ohmic contact between the bulk heterojunction (BHJ) and the cathode. The synergistic improvement of these interfacial properties is proven to effectively suppress nongeminate recombination and enhance charge extraction efficiency within the device. Therefore, the use of the PFN-Br:MA composite interlayer in our work is intended to leverage this synergistic mechanism to optimize interfacial properties for improved overall device performance. We have cited this reference in the revised manuscript.

7. Since the authors reported a very high PCE, I suggest the value to be examined by the NIM, China.

Reply: Thank you for the valuable suggestions. The National Institute of Metrology (NIM) is a highly authoritative certification center with a substantial workload, making

appointments there quite difficult. We sincerely apologize for not being able to obtain efficiency certification from NIM this time.

The reliability of South China National Center of Metrology (GuangDong Institute of Metrology), abbreviating SCM, is grounded in the following aspects: (1) it is one of the seven major regional national legal metrological verification institutions in China, a non-profit organization established by the government, which ensures the authority and impartiality of its certificates and reports. (2) the center is accredited by the China National Accreditation Service for Conformity Assessment (CNAS), with testing capabilities covering optical and related fields. It is also a CNAS-accredited proficiency testing provider. (3) the center has established several national and regional metrological standards and is equipped with internationally advanced testing instruments, ensuring that all measurements are traceable to national standards. In line with the principle of geographical convenience, many research institutions in South China and neighboring regions choose this center for photovoltaic efficiency certification. Relevant results have been published in high-level journals such as *Science*, **2025**, 390, 905-910; *Nat. Synth.* **2025**, <https://doi.org/10.1038/s44160-025-00904-6>; *Adv. Mater.* 2025, 37, 2411989; *Adv. Mater.* 2025, 37, e09806, further attesting to its recognition within the academic community.

8. In Page 14, the authors expressed the significant slowdown of charge recombination. what is the basis?

Reply: We thank you for your careful reading and comment. From the fs-TAS spectra of the D18:S-Cb:L8-BO blend films with different L8-BO ratios (Fig. 2d), it can be observed that after 100 ps (the period where charge recombination generally becomes dominant), the decay trend of the signal for the D18:S-Cb blend film (green curve, 0% L8-BO) is slower (longer carrier lifetime) than that of the D18:L8-BO blend film (red curve, 100% L8-BO), which suggests that charge recombination in the D18:S-Cb blend is slower, indicating that compared to the D18:L8-BO blend, charge recombination in the D18:S-Cb blend is somewhat slower.

We have revised the manuscript accordingly:

In contrast, the charge recombination process in D18:S-Cb slows down, as indicated by its more gradual decay trend (green curve, 0% L8-BO), while the D18:L8-BO exhibits a faster decay after 100 ps (red curve, 100% L8-BO), which is beneficial for achieving higher EQE and FF in devices.

9. Why does the larger LUMO-LUMO energy offset induce the larger ΔE_2 ? The authors should reveal under which conditions the energy losses were evaluated, CT or S-Q theory.

Reply: Thank you for the insightful comment. In OSCs, energy loss (E_{loss}) is commonly categorized into three components: ΔE_1 , ΔE_2 , and ΔE_3 . Among these, ΔE_1 is regarded as an inevitable intrinsic loss, primarily stemming from radiative recombination associated with photon absorption above the bandgap of the active layer materials. Its magnitude generally shows a positive correlation with the blend film's bandgap. ΔE_2 is defined as an additional radiative loss induced by sub-bandgap absorption, a process closely linked to the driving force for charge separation. ΔE_3 corresponds to non-radiative recombination losses, typically caused by factors such as defects and vibronic coupling. In the prevailing non-fullerene acceptor systems, the energy level offset between LUMO of the donor and acceptor (ΔE_{LUMO}) is usually significantly larger than the offset between their HOMO (ΔE_{HOMO}). Consequently, the generation of ΔE_2 can be primarily attributed to the electron transfer process from the LUMO of donor (LUMO^{D}) to the LUMO of acceptor (LUMO^{A}). From a physical perspective, the ΔE_{LUMO} value can be approximately correlated with the energy difference between the donor's local excited state (LE^{D} state) and the charge-transfer state (CT state), as schematically illustrated in **Fig. R4**. The magnitude of ΔE_2 fundamentally reflects the electronic coupling strength between the CT and LE states: stronger coupling typically leads to a larger ΔE_2 . When the LUMO level of the acceptor is lowered (i.e., ΔE_{LUMO} increases), the electron distribution becomes more localized on the CT state. This reduced charge delocalization enhances the coupling between the LE and CT states. According to

relevant research (e.g., *Nat. Mater.* 2018, 17, 703-709), the mechanism described above suggests that an excessively large donor-acceptor LUMO-LUMO energy offset is often a key factor contributing to an increased ΔE_2 . In this work, our analysis of energy loss for D18:S-Cb:L8-BO ternary system was conducted primarily within the Shockley-Queisser (S-Q) theoretical framework. We have emphasized this in the revised manuscript.

We have revised the manuscript accordingly:

The Fourier transform photonic spectroscopy EQE (**Supplementary Fig. 18**) and electroluminescence (EL) spectra (**Fig. 2f**) of D18:S-Cb:L8-BO system were measured to assess E_{loss} of OSCs based on Shockley-Queisser (S-Q) theoretical framework.

Fig. R4 Schematic diagram of LUMO, LE state and CT state.

10. The authors should define the δG , what is physical meaning of the term?

Reply: We sincerely appreciate the valuable suggestion. We have carefully revised and add δG explanation in SI.

In the independent gradient model based on Hirshfeld partition (IGMH) analysis (*J. Comput. Chem.*, **43**, 539-555 (2022)), the atomic densities involved in definition of δg is derived based on Hirshfeld partition, namely

$$\rho_i^{\text{Hirsh}} = \rho(\mathbf{r})w_i(\mathbf{r})$$

where ρ is the electron density of the whole system calculated based on wavefunction, and the Hirshfeld weighting function of atom i is expressed as

$$w_i(\mathbf{r}) = \frac{\rho_i^{free}(\mathbf{r})}{\rho^{pro}(\mathbf{r})} = \frac{\rho_i^{free}(\mathbf{r})}{\sum_j \rho^{pro}(\mathbf{r})}$$

where ρ_i^{free} is spherically averaged electron density of atom i in its free state, ρ^{pro} corresponds to promolecular density, the index j loops over all atoms.

For three-dimensional cases, g^{IGM} and δg can be defined as follows.

$$g(\mathbf{r}) = \left| \sum_i \nabla \rho_i^{free}(\mathbf{r}) \right|$$

$$g^{IGM}(\mathbf{r}) = \sum_i |\nabla \rho_i^{free}(\mathbf{r})|$$

$$\delta g(\mathbf{r}) = g^{IGM}(\mathbf{r}) - g(\mathbf{r})$$

The ρ_i^{free} stands for spherically averaged density of atom i in its free state.

The atomic pair δg index (δG_{pair}) was defined to quantify the contribution of atomic pair to the interaction between two fragments (A and B)

$$\delta G_{i,j}^{pair} = \int \delta g_{i,j}(\mathbf{r}) d\mathbf{r} = \int [g_{i,j}^{IGM}(\mathbf{r}) - g_{i,j}(\mathbf{r})] d\mathbf{r} \quad i \in A, j \in B$$

where

$$g_{i,j}(\mathbf{r}) = |\nabla \rho_i^{free}(\mathbf{r}) + \nabla \rho_j^{free}(\mathbf{r})|$$

$$g_{i,j}^{IGM}(\mathbf{r}) = |\nabla \rho_i^{free}(\mathbf{r})| + |\nabla \rho_j^{free}(\mathbf{r})|$$

It is also useful to define percentage atomic pair contribution to interfragmentary interaction as

$$\delta G_{i,j}^{pair}(\%) = \frac{\delta G_{i,j}^{pair}}{\sum_{k \in A} \sum_{l \in B} \delta G_{k,l}^{pair}}$$

It is able to accurately represent contribution of atomic pairs to interaction energy between two fragments, however $\delta G_{pair}(\%)$ should be able to identify “hot” atomic pairs, which may indeed have large actual contribution to interfragmentary binding.

11. The clamping effect in Figure 4d and 4e is not indicated clearly, and I suggest the authors to redraw the picture to clearly show the clamping dimer geometry. Especially, the hydrogen bonding should be clearly indicated and confirmed with solid evidence.

Reply: We thank you for this excellent suggestion. We agree that the original figures did not optimally illustrate the intricate geometry of the clamp dimer and the critical hydrogen bonding interactions. Following the your advice, we have redesigned and redrawn **Fig. 4d** to provide a much clearer and more informative visualization.

Fig. 4 | Molecular dynamics simulations. (d) Weak interaction in clamp-type dimer.

We now present the clamp dimer from clear viewing angles to show how the planar cyclobutyl group of S-Cb inserts into the bifurcated pocket of the L8-BO side chain. The key atoms involved in the hydrogen bonding (C-H \cdots H-C interactions) are explicitly shown to emphasize their spatial proximity. The hydrogen bonds are now unambiguously depicted as dashed lines connecting the cyclobutyl group and bifurcated atoms, clearly labeled with the calculated interatomic distances. The distances between the hydrogen atoms on the S-Cb cyclobutyl ring and the L8-BO branched chain are consistently found to be in the range of 2.2 - 2.8 Å.

12. For the PiFM, why 1426 cm⁻¹ is choosed for S-Cb and 1532 cm⁻¹ for L8-BO. Are they selected for the specific functional groups?

Reply: We thank you for the valuable comments. The working principle of PiFM is that when the wavelength of the tunable infrared laser resonates with the vibrational frequency of a local chemical bond in the sample, an induced dipole is excited. A detectable photo-induced force arises between this dipole and the image dipole of the metallized AFM tip. PiFM images the sample by probing this photoelectric force, and its signal intensity directly reflects the sample's light absorption capacity at a specific wavelength. Although S-Cb and L8-BO share similar chemical structures, slight differences in their alkyl chains and molecular backbone torsion lead to precise shifts in the positions of their characteristic infrared absorption peaks. The core principle guiding our choice of imaging wavenumbers is to select the position where the target component's signal is strongest while ensuring minimal interfering signal from the other component. For the S-Cb component (Characteristic peak: 1426 cm⁻¹): This wavenumber corresponds to the position of the strongest absorption band for the C-H bending vibration on the alkyl chains of the S-Cb molecule. Although L8-BO also has C-H bending vibrations in a similar region, its strongest peak is located at 1402 cm⁻¹. Consequently, excitation at 1426 cm⁻¹ elicits the strongest photo-induced force response from S-Cb, while the signal from L8-BO is weak and does not constitute substantial interference for imaging the S-Cb morphology. For the L8-BO component

(Characteristic peak: 1532 cm^{-1}): This wavenumber corresponds to the position of the strongest absorption band for the C=C stretching vibration on the aromatic backbone of the L8-BO molecule. Due to the aforementioned conformational differences, the analogous vibrational peak for S-Cb is shifted to 1501 cm^{-1} . Therefore, imaging at 1532 cm^{-1} allows for the specific, high-contrast mapping of L8-BO distribution, with negligible signal interference from S-Cb. Our selection of 1426 cm^{-1} and 1532 cm^{-1} as the characteristic imaging wavenumbers for S-Cb and L8-BO, respectively, is not based on unique functional groups exclusive to each (such as the carbonyl group $\sim 1700\text{ cm}^{-1}$ or the cyano group $\sim 2200\text{ cm}^{-1}$). Instead, it is based on the precise utilization of the peak position shifts within their shared vibrational modes (alkyl C-H bending and aromatic C=C stretching), induced by subtle structural tuning. This strategy ensures that during PiFM characterization of the ternary blend film (D18:S-Cb:L8-BO), the spatial distribution of the two structurally similar acceptor components can be clearly and reliably resolved without significant cross-talk between their signals.

13. The authors demonstrate that ternary system shows better morphological stability under high temperature. I think it is necessary to examine and discuss the stability of OPV devices and make a comparison.

Reply: Thank you for the valuable suggestions. We have tested the thermal- and photo-stability of devices based on D18:S-Cb:L8-BO with L8-BO contents of 0%, 50%, and 100%, as shown in **Supplementary Fig. 33**. The results indicate that although the overall thermal- and photo- stability of the ternary devices remains inferior to state-of-the-art values reported in the literature—particularly in terms of thermal stability—these results provide indirect evidence supporting the role of the hydrogen-bond-guided clamping effect in modulating blend morphology and device stability. We will continue to focus on and optimize the device performance under long-term operating conditions.

We have added the data and discussion in the manuscript accordingly:

To further investigate the impact of the clamping effect on thermal- and photo-stability of OSCs, we fabricated inverted devices with a structure of ITO/ZnO/active

layer/MoO_x/Ag. The stability test results for devices based on D18:S-Cb:L8-BO with 0%, 50%, and 100% L8-BO content are presented in **Supplementary Fig. 33**. Under continuous annealing at 100 °C, D18:S-Cb device exhibited significantly better thermal stability than the D18:L8-BO device, which can be attributed to the higher T_g of S-Cb resulting from its short and rigid cyclobutyl substituent. Furthermore, when 50% L8-BO was incorporated into the D18:S-Cb active layer, the device stability showed improvement within the first 160 h of heating. However, beyond 160 h, the thermal stability of the ternary device declined more rapidly compared to the D18:S-Cb binary device. We speculate that during the early stage of heating, the clamp effect mediated by hydrogen bonding helps to stabilize the morphology. With prolonged thermal stress, this hydrogen-bond network may gradually degrade, causing the ternary blend to evolve toward a morphology resembling that of D18:L8-BO, thus accelerating the decay in device performance.

It is worth noting that, under continuous heating at 100 °C, these weak interactions provide limited stabilization, which explains why the stability improvement is modest and eventually deteriorates over extended periods, leading to the observed trend of initial stability followed by rapid decay in the ternary system. In contrast, under continuous illumination, such hydrogen-bonding interactions appear less susceptible to degradation. Consequently, the device based on D18:S-Cb:L8-BO with 50% L8-BO content demonstrated markedly better photo-stability than the D18:S-Cb binary device. Although the overall thermal- and photo- stability of the ternary devices remains inferior to state-of-the-art values reported in the literature⁵⁶—particularly in terms of thermal stability—these results provide indirect evidence supporting the role of the hydrogen-bond-guided clamping effect in modulating blend morphology and device stability.

Supplementary Fig. 33 (a) Thermal- and (b) photo-stability test of OSCs based on D18:L8-BO, D18:S-Cb and D18:S-CbLL8-BO (0.5:0.5).

14. Traditionally, the alloy mechanism was characterized by the linear correlation between V_{oc} and the ratio of the third component.

Reply: Thank you for the helpful suggestion. we plotted V_{oc} as a function of L8-BO proportion (**Supplementary Fig. 19**). The results show that the V_{oc} of the D18:S-Cb:L8-BO device increases in a well-defined linear manner with increasing L8-BO content. This linear dependency aligns closely with the characteristic behavior of an alloy model, suggesting that S-Cb and L8-BO form an alloy-like composite phase, thereby supporting an alloy-type working mechanism.

We have added it into the revised manuscript accordingly:

As L8-BO content gradually increases, the V_{oc} of D18:S-Cb:L8-BO-based ternary devices exhibits a linear improvement (**Supplementary Fig. 19**), suggesting that the potential mechanism for improved performance of ternary devices may be the formation of acceptor alloy phase.

Reviewer #3:

This study reports a novel cyclobutyl-substituted small-molecule acceptor (S-Cb) that forms interchain supramolecular “clamp” interactions with L8-BO, effectively suppressing electron-phonon coupling and enabling ternary organic solar cells (OSCs)

with an impressive efficiency of 20.93% (certified 20.74%). This work is highly innovative, systematic, and thorough. The molecular design is rational, the experimental support is comprehensive, and the combination of morphology characterization, device physics, and theoretical simulations provides a convincing mechanism for the performance enhancement. The reported efficiency is among the highest in the field of OSCs. This manuscript presents a timely and impactful contribution to the field of high-efficiency OSCs. The clamp strategy offers a new pathway to suppress electron-phonon coupling and optimize active layer morphology. The work meets the high standards of Nature communications and is recommended for acceptance after minor revisions.

1. The introduction of a rigid cyclobutyl side chain to enhance molecular rigidity and suppress electron-phonon coupling is a clever and underexplored strategy, Compare S-Cb with cyclopentyl/cyclohexyl analogues to highlight ring-strain superiority. "Clamping effect" requires clearer definition (e.g., schematic illustrating H-bond positioning).

Reply: We sincerely thank you for your valuable suggestion. To better elucidate the unique advantages arising from the ring strain of the cyclobutyl group, we systematically compared the structure properties of common cyclic alkyl chains—namely cyclohexyl, cyclopentyl, cyclobutyl, and cyclopropyl—in our study. This comparative analysis was conducted mainly for the following reasons: on one hand, it helps to clearly illustrate the structure–property relationship of the cyclobutyl group in a principled manner; on the other hand, given that the substitution positions and specific structures of cyclic alkyl chains reported in the literature vary considerably, a direct cross-literature comparison of properties could easily be confounded by structural inconsistencies. Therefore, we chose to perform the comparison within a controlled and consistent framework, in order to more clearly delineate the distinctive role of the cyclobutyl group.

We have added the discussion into the revised manuscript accordingly:

Among common cyclic alkyl chains, such as cyclohexyl, cyclopentyl, cyclobutyl and cyclopropyl, the angles of C-C bond are approximately 109° , 108° , 90° , and 60° , respectively. Cyclohexyl and cyclopentyl groups can largely maintain the ideal tetrahedral angle of 109.5° for sp^3 -hybridized carbon atoms, resulting in low ring strain. These rings adopt characteristic conformations—the non-planar “envelope” conformation for cyclopentane and the stable chair conformation for cyclohexane. In contrast, cyclopropyl exhibits the highest ring strain and a rigid planar geometry. Although its pronounced ring strain could, in principle, enhance structural rigidity, the cyclopropyl ring is prone to ring-opening reactions under various chemical conditions, including those encountered during the synthesis of Y6-type derivatives, which limits its practical utility in such materials. Cyclobutyl with a C-C bond angle of about 90° that significantly deviates from the ideal tetrahedral angle possesses considerable ring strain that imparts notable conformational rigidity. To partially relieve this strain, the cyclobutane ring adopts a slightly folded “butterfly” conformation, with a dihedral angle between the ring planes of only about 14° , indicating a relatively flat structure. This combination of substantial rigidity, moderate steric bulk, and sufficient synthetic stability makes the cyclobutyl group an attractive structural motif in the design of organic photovoltaic materials. Moreover, the structural characteristics of cyclobutane place it between flexible alkyl chains and rigid aromatic groups, ensuring molecular solubility while imparting moderate rigidity to the alkyl chain region.

Moreover, we have clearly defined the “clamping effect” in the revised manuscript and have correspondingly indicated the relevant hydrogen-bonding sites in **Fig. 4d** to provide a clearer visual illustration of this structural feature.

Fig. 4 | Molecular dynamics simulations. (d) Weak interaction in clamp-type dimer.

We have added this into the revised manuscript accordingly:

the clamp effect (refers to a supramolecular interaction wherein the nearly planar and rigid cyclobutyl side chain of S-Cb is spatially locked into the branched alkyl chain pocket of L8-BO via intermolecular hydrogen bonds (e.g., C-H \cdots H-C), effectively

“clamping” the side chains of the two acceptors together) is most obvious

2. The "clamping effect" between S-Cb and L8-BO is convincingly demonstrated through GIWAXS, PiFM, NMR, and molecular dynamics simulations. Although NMR shows spatial proximity between S-Cb and L8-BO. FTIR or temperature-dependent NMR could further confirm the presence and strength of hydrogen bonding.

Reply: Thank you for your valuable suggestion. we have performed the Fourier-transform infrared (FTIR) spectroscopy on neat films of S-Cb and L8-BO, as well as their blend film with a weight ratio of 1:1. The corresponding spectra are provided in **Supplementary Fig. 28**. By comparing the FTIR transmission spectra of the neat S-Cb and L8-BO films, we assigned the characteristic C-H stretching vibration of the cyclobutyl group at approximately 2954 cm^{-1} . Notably, in the S-Cb:L8-BO blend film, this C-H vibration peak exhibits a small but discernible shift to a higher wavenumber (blue-shift of about 2 cm^{-1}) compared to the neat S-Cb film, accompanied by an increase in peak intensity and broadening of the line shape. These spectral changes align with the trend expected when the cyclobutyl C-H bond acts as a potential hydrogen-bond acceptor. We consider that the relatively small magnitude of the shift may be attributed to the moderate polarity of the cyclobutyl C-H bond, which could limit the strength of its hydrogen-bond accepting interaction, thereby resulting in only subtle yet detectable alterations in the infrared spectrum.

We have added the data and discussion into the revised manuscript accordingly:

To further prove the existence of C-H \cdots H-C hydrogen bond, we have performed the Fourier-transform infrared (FTIR) spectroscopy on neat films of S-Cb and L8-BO, as well as their blend film with a weight ratio of 1:1, corresponding spectra are provided in **Supplementary Fig. 28**. By comparing the FTIR transmission spectra of the neat S-Cb and L8-BO films, we assigned the characteristic C-H stretching vibration of the cyclobutyl group at approximately 2954 cm^{-1} . Notably, in the S-Cb:L8-BO blend film, this C-H vibration peak exhibits a small but discernible shift to a higher wavenumber (blue-shift of about 2 cm^{-1}) compared to the neat S-Cb film, accompanied by an

increase in peak intensity and broadening of the line shape. These spectral changes align with the trend expected when the cyclobutyl C-H bond acts as a potential hydrogen-bond acceptor. It is worth noting that the considerable strain and the resulting bent bonds enhance the polarity of the C-H bonds on cyclobutyl, endowing the hydrogen atoms with a partial positive charge-though still weaker than that of hydrogens bonded directly to oxygen or nitrogen. The compact steric profile and the weakly electrophilic hydrogen of the cyclobutyl group allow it to act as a hydrogen-bond acceptor, forming weak but non-negligible hydrogen bonds. We consider that the relatively small magnitude of the shift may be attributed to the moderate polarity of the cyclobutyl C-H bond, which could limit the strength of its hydrogen-bond accepting interaction, thereby resulting in only subtle yet detectable alterations in the FTIR spectrum.

3. Quantification of Electron-Phonon Coupling is required. While Stokes shift, FWHM, and Huang-Rhys factors are provided, temperature-dependent PL or charge transport measurements could help quantify the electron-phonon coupling constant more directly.

Reply: Thank you for the valuable suggestion. We have conducted temperature-dependent photoluminescence (PL) measurements on L8-BO, S-Cb neat films, and their blended film (S-Cb:L8-BO with a weight ratio of 1:1). The temperature was varied from 300 K down to 80 K in steps of 20 K. The corresponding results are presented in **Supplementary Fig. 25**. To quantitatively evaluate the electron-phonon coupling strength, we applied a single effective phonon mode and performed an approximate linear fitting to the linewidth broadening within the temperature range of 200 K to 80 K. The extracted electron-phonon coupling constants are approximately 0.305 meV K^{-1} for L8-BO, 0.284 meV K^{-1} for S-Cb, and 0.163 meV K^{-1} for the S-Cb:L8-BO blend. The fitting results indicate that the electron-phonon coupling in neat S-Cb is slightly weaker than that in neat L8-BO. More importantly, the blending of the two materials at a 1:1 ratio leads to a more pronounced suppression of the coupling. This experimental trend is consistent with our theoretical calculations. We attribute this enhanced

suppression to the formation of L8-BO-S-Cb dimer via hydrogen-bonding interactions between the side chains of S-Cb and L8-BO, which likely helps to reduce vibrational disorder.

We have added the data and discussion into the revised manuscript accordingly:

To quantitatively evaluate the electron-phonon coupling strength, temperature-dependent PL measurements were performed on L8-BO, S-Cb neat films, and their blended film with a 1:1 weight ratio. The temperature was varied from 300 K down to 80 K in steps of 20 K. The PL spectra and corresponding fitting results are provided in **Supplementary Fig. 25**. The broadening of the PL emission linewidth with increasing temperature can generally be described by the following model^{19,54,55}:

$$\Gamma(T) = \Gamma_i + a \exp\left(-\frac{Ea}{k_b T}\right) + bT$$

where, Γ_i represents the inhomogeneous linewidth of the film, a denotes the density of non-radiative recombination centers, Ea is regarded as the energy barrier for back charge recombination from electronic to excitonic states in Y-series non-fullerene acceptors, and b is defined as the electron-phonon coupling coefficient. In the low-temperature region (80-200 K), the contribution of the second (nonlinear) term to the linewidth is minor; therefore, an approximate linear fitting was applied to data in this range to extract the electron-phonon coupling constant b . The fitting yields b values of approximately 0.305, 0.284, and 0.163 meV·K⁻¹ for neat L8-BO, S-Cb, and the S-Cb:L8-BO blend, respectively. These results indicate that the electron-phonon coupling in neat S-Cb is slightly weaker than that in neat L8-BO. Notably, blending the two materials at a 1:1 weight ratio leads to a more pronounced suppression of the coupling, a trend consistent with our theoretical calculations. We attribute this enhanced suppression to the formation of larger dimer-like structures via hydrogen-bonding interactions between the side chains of S-Cb and L8-BO, which likely further restrains molecular vibrations.

Supplementary Fig. 25 Temperature-dependent PL spectra of (a) L8-BO neat film, (b) S-Cb neat film, and (c) S-Cb:L8-BO (1:1) blend film. (d) Linear fitting in the low-temperature region between temperature and FWHM.

4. The morphological stability under thermal stress (100°C) is shown in Supplementary Fig. 22, but light-soaking stability should be included to evaluate operational stability.

Reply: We sincerely appreciate your valuable suggestions. In response to your suggestions, we have conducted tests on the thermal and photo-stability of OSCs based on D18:S-Cb:L8-BO system, with L8-BO contents of 0%, 50%, and 100%, respectively. The corresponding results are provided in **Supplementary Fig. 33**.

We have added the data and discussion into the revised manuscript accordingly:

To further investigate the impact of the clamping effect on thermal- and photo-stability of OSCs, we fabricated inverted devices with a structure of ITO/ZnO/active layer/MoO_x/Ag. The stability test results for devices based on D18:S-Cb:L8-BO with 0%, 50%, and 100% L8-BO content are presented in **Supplementary Fig. 33**. Under continuous annealing at 100 °C, D18:S-Cb device exhibited significantly better thermal

stability than the D18:L8-BO device, which can be attributed to the higher T_g of S-Cb resulting from its short and rigid cyclobutyl substituent. Furthermore, when 50% L8-BO was incorporated into the D18:S-Cb active layer, the device stability showed improvement within the first 160 h of heating. However, beyond 160 h, the thermal stability of the ternary device declined more rapidly compared to the D18:S-Cb binary device. We speculate that during the early stage of heating, the clamp effect mediated by hydrogen bonding helps to stabilize the morphology. With prolonged thermal stress, this hydrogen-bond network may gradually degrade, causing the ternary blend to evolve toward a morphology resembling that of D18:L8-BO, thus accelerating the decay in device performance.

It is worth noting that, under continuous heating at 100 °C, these weak interactions provide limited stabilization, which explains why the stability improvement is modest and eventually deteriorates over extended periods, leading to the observed trend of initial stability followed by rapid decay in the ternary system. In contrast, under continuous illumination, such hydrogen-bonding interactions appear less susceptible to degradation. Consequently, the device based on D18:S-Cb:L8-BO with 50% L8-BO content demonstrated markedly better photo-stability than the D18:S-Cb binary device. Although the overall thermal- and photo- stability of the ternary devices remains inferior to state-of-the-art values reported in the literature⁵⁶—particularly in terms of thermal stability—these results provide indirect evidence supporting the role of the hydrogen-bond-guided clamping effect in modulating blend morphology and device stability.

Supplementary Fig. 33 (a) Thermal- and (b) photo-stability test of OSCs based on D18:L8-BO, D18:S-Cb and D18:S-CbLL8-BO (0.5:0.5).

5. A PCE of 20.93% (certified 20.74%) with low non-radiative loss and good reproducibility represents a state-of-the-art result. The non-radiative loss (ΔE_3) increases again at 75% L8-BO content. Please discuss whether this is related to morphological deterioration or reduced clamp efficiency.

Reply: Thank you for the insightful comments. In the ternary OSCs based on the D18:S-Cb:L8-BO system, the ΔE_3 exhibits a trend of initial decrease followed by an increase with the rising L8-BO content. This variation is consistent with the proposed “clamping effect”, which first strengthens and then weakens. Specifically, the clamping effect is the weakest at an L8-BO content of 75%. According to the theoretical calculations, this clamping effect can reduce both the Huang–Rhys factor and the free volume of the active layer. Consequently, the attenuation of the clamping effect at the 75% L8-BO content is identified as a key factor leading to the subsequent increase in ΔE_3 .

We have added the data and discussion into the revised manuscript accordingly:

However, when L8-BO content is increased to 75%, the EQE_{EEL} of D18:S-Cb:L8-BO decreases, indicating an increase in ΔE_3 (the reason will be discussed in the theoretical calculation section).

The clamping effect helps to reduce both the Huang-Rhys factor and the free volume of ternary active layer, thereby effectively suppressing electron-phonon coupling to improve EQE_{EEL} of D18:S-Cb:L8-BO ternary active blend and suppress ΔE_3 ²¹.

6. Alloy-phase evidence should be consolidated. Specify if "acceptor alloy" implies mixed crystalline/amorphous phases. CV shows linear LUMO shift; DSC shows single crystallization peak. However, GIWAXS (100) peak drift is $<0.04 \text{ \AA}$, much smaller than high-quality alloys ($>0.1 \text{ \AA}$). Statistical significance (error bars, x parameter) is absent.

MD g(r) alone is insufficient; orientational order parameter P2 and Flory-Huggins X should be included to demonstrate thermodynamic miscibility.

Reply: Thank you for the valuable comments and insightful suggestions. We calculated the positions of the π - π stacking (010) peak in the D18:S-Cb:L8-BO blend films with varying L8-BO contents (25%, 50%, 75%). The peak drifts relative to the D18:S-Cb binary film are 0.046 Å, 0.061 Å, and 0.042 Å, respectively. The data trend indicates that the acceptor alloy phase achieves its highest relative degree of order at the 50% L8-BO ratio. We appreciate the reviewer's attention to the numerical difference compared to some highly ordered alloy phases reported in the literature (where peak drifts often exceed 0.1 Å). We believe the degree of order of S-Cb:L8-BO alloy phase may differ from the (010) peak drift shown by highly ordered alloy phases reported in the literature. Our judgment regarding the formation of an acceptor alloy phase between S-Cb and L8-BO is primarily based on the following three lines of experimental evidence: (1) The V_{OC} of the ternary devices based on D18:S-Cb:L8-BO shows a well-defined linear increase with the content of the higher- V_{OC} component, L8-BO, which is consistent with the operational mechanism of an acceptor alloy phase. (2) The LUMO and HOMO energy levels of the S-Cb:L8-BO blend films exhibit continuous, linear shifts with the L8-BO ratio, indicating good electronic state mixing at the molecular level. (3) DSC measurements reveal a single endothermic melting peak for the S-Cb:L8-BO mixture, with a melting point lower than that of the S-Cb or L8-BO component. This supports the formation of a single alloy phase from a thermodynamic perspective. Furthermore, the high structural similarity between S-Cb and L8-BO provides a favorable foundation for the formation of a homogeneous acceptor alloy phase, based on the principle of "like dissolves like." Regarding the observed (010) peak drift values (0.061 Å) not exceeding 0.1 Å in our system, we consider that this may be related to the specific molecular packing behavior and complex intermolecular interactions within this multicomponent system. Factors such as the inherent packing energetics of the acceptor molecules themselves and the potential influence of the

presence of the donor material D18 on the microstructure of the acceptor phase could modulate the final order degree of the alloy phase.

Furthermore, we have performed additional contact angle measurements using ethylene glycol and water on the neat films of both acceptors in order to thermodynamically evaluate the ease with which S-Cb and L8-BO form an acceptor alloy phase. These tests were conducted to derive their surface tension and estimate the Flory-Huggins interaction parameter (χ) between them. The corresponding results are compiled and presented in **Supplementary Fig. 20**. Analysis of the data shows that the surface tension values of L8-BO and S-Cb are very similar, indicating a relatively small Flory-Huggins interaction parameter (χ) between the two components. A small χ value suggests low incompatibility upon mixing, which thermodynamically supports the favorable formation of a homogeneous acceptor alloy phase between S-Cb and L8-BO.

We have added the data and discussion into the revised manuscript accordingly:

To thermodynamically probe the ease of forming an acceptor alloy phase between S-Cb and L8-BO, we evaluated their compatibility by measuring the contact angles of their neat films and deriving the corresponding surface tensions. By using ethylene glycol (EG) and water as probe liquids (contact angles of 65.40°/94.19° for L8-BO and 61.66°/89.77° for S-Cb, respectively, **Supplementary Fig. 20**), the surface tensions (λ) of L8-BO and S-Cb were fitted via Wu's model to be 39.68 mN/m and 40.33 mN/m, respectively. The Flory-Huggins interaction parameter χ , estimated from $\chi \propto (\sqrt{\lambda_{S-Cb}} - \sqrt{\lambda_{L8-BO}})^2$ was found to be exceptionally small ($\chi \propto 0.0026$). This minimal χ value indicates excellent thermodynamic miscibility between S-Cb and L8-BO, providing a key thermodynamic rationale for their facile formation of a homogeneous acceptor alloy phase in the blend.

Supplementary Fig. 20 Contact angle measurement of L8-BO and S-Cb neat films.

7. Synthetic Scalability: Cyclobutyl synthesis complexity may hinder industrial adoption. A brief cost/feasibility assessment is needed. Industrial viability (e.g., ink formulation, large-area printing compatibility) should be discussed.

Reply: We sincerely thank you for the valuable suggestions. We compared the cost of S-Cb with that of L8-BO, the most advanced electron acceptor currently available. The key to scaling up the production of L8-BO lies in the high-yield synthesis of the 3-(2-butyloctyl)thieno[3,2-*b*]thiophene monomer. Generally, there are three common synthetic routes, as shown below:

Fig. R1 Synthetic route **1** for 3-(2-butyl-octyl)thieno[3,2-*b*]thiophene.

Fig. R2 Synthetic route **2** for 3-(2-butyl-octyl)thieno[3,2-*b*]thiophene.

Fig. R3. Synthetic route **3** for 3-(2-butyl-octyl)thieno[3,2-*b*]thiophene.

In synthetic route **1**, reagents such as PCC (containing heavy metal Cr^{3+}), diethyl ether (highly explosive), and lithium aluminum hydride (flammable) are required, and the process involves up to 7 steps from the starting material to the target compound. Moreover, the yield of the dehydroxylation step (below) in this route is difficult to control consistently.

Therefore, route **1** may face significant challenges for future large-scale production.

In synthetic route **2**, when the reaction scale exceeds 10 grams, the yield of the step (below) decreases considerably.

However, by employing the reaction conditions corresponding to synthetic route 3, the yield of this step (below) can exceed 90%.

We have successfully scaled up the synthesis to 50 grams with maintained yield. Based on our experimental experience, route 3 (currently used for synthesizing S-Cb) appears to be the most feasible approach for the industrial-scale synthesis of 3-(2-butyloctyl)thieno[3,2-*b*]thiophene in the future.

When preparing L8-BO and S-Cb via route 3, the only difference in starting materials is that L8-BO requires 2-butyloctyl bromide (Br-BO), while S-Cb requires (bromomethyl)cyclobutene (Br-Cb). Currently, the market price of Br-BO (CAS No.: 85531-02-8) is about \$3.2 per gram, whereas Br-Cb (CAS No.: 17247-58-4) costs approximately \$1.1 per gram (the alkyl chain of Br-Cb is commercially available in large quantities). Additionally, considering that S-Cb uses a BO chain as the alkyl group linked to the nitrogen atom in the subsequent synthesis route, while L8-BO employs a 2-ethylhexyl (EH) chain, the overall cost of the alkyl chain combinations in both materials is comparable. Given that the current market price of L8-BO is around \$450 per gram, we estimate that S-Cb would be priced at a similar level.

Regarding the compatibility of S-Cb with inkjet printing and large-area coating techniques, further investigation is still needed. However, we preliminarily believe that the existing printing and coating methods suitable for Y6 and L8-BO are likely to be applicable to S-Cb as well.

We have added the discussion into the revised manuscript accordingly:

Evaluation reveals that if L8-BO is synthesized via the same route as S-Cb, the alkyl chain combinations in both materials would lead to comparable large-scale production costs for S-Cb relative to L8-BO. It is worth noting that (bromomethyl)cyclobutene (the

carboxylic acid precursor containing one extra carbon atom) required for S-Cb is commercially available at a lower price than the 2-butyloctyl bromide used for L8-BO.

Reviewers' Comments to Authors:

Reviewer #2:

The authors provided extensive revisions to the manuscript and a comprehensive response to my previous comments. The efforts to clarify the scientific content and to elaborate on the material design are evident and have improved the manuscript. However, there remains some concerns need to be addressed.

1) There is some conceptual confusion regarding the definitions of radiative energy loss (ΔE_2) and non-radiative energy loss (ΔE_3). Both ΔE_2 and ΔE_3 are linked to the energy offset between the donor and the acceptor. However, in the authors' definitions, ΔE_2 is solely ascribed to this energy offset, whereas ΔE_3 is attributed to other factors such as defects and vibronic coupling. Furthermore, it should be noted that in the current donor-acceptor system, the key factor influencing radiative/non-radiative recombination should be the HOMO offset, rather than LUMO offset. Additionally, Fig. R4 shows that the CT energy level equals the LUMO energy level of the acceptor, and the LE energy level equals the LUMO energy level of the donor. What is the basis for this assumption?

Reply: We are very grateful to you for recognizing our efforts to improve the quality of the manuscript, and for your critical and insightful comment again.

For organic solar cells (OSCs), energy loss (E_{loss}) is defined and consists of three parts, as follows:

$$\begin{aligned}
 E_{loss} &= E_{gap} - qV_{OC} = (E_{gap} - qV_{OC}^{SQ}) + (qV_{OC}^{SQ} - qV_{OC}^{Rad}) + (qV_{OC}^{Rad} - qV_{OC}) \\
 &= (E_{gap} - qV_{OC}^{SQ}) + q \Delta V_{OC}^{Rad, below gap} + q \Delta V_{OC}^{Non-rad} \\
 &= \Delta E_1 + \Delta E_2 + \Delta E_3
 \end{aligned}$$

Where V_{OC}^{SQ} in the equation is the maximum voltage based on the Shockley-Queisser limit. The first term of energy loss in equation ($\Delta E_1 = E_{gap} - qV_{OC}^{SQ}$) is due to the mismatch between radiation received in a narrow solid angle from the sun and omnidirectional radiative recombination originating from the absorption above the

bandgap. This loss is unavoidable for any type of solar cells and is typically 0.25 eV or above for OSCs. The second term in the equation ($\Delta E_2 = q \Delta V_{OC}^{Rad, below\ gap}$) is due to additional radiative recombination from the absorption below the bandgap. The third loss term ($\Delta E_3 = q \Delta V_{OC}^{Non-rad} = -kT \ln(EQE_{EL})$) is due to the non-radiative recombination, where EQE_{EL} is electroluminescence quantum efficiency of the solar cell when charge carriers are injected into the device in dark.

From the definition and composition of E_{loss} , ΔE_2 is defined as the additional radiative energy loss arising from absorption below the bandgap. Accordingly, in order to facilitate the differentiation of ΔE_1 , we have revised the description of ΔE_2 in the manuscript to “additional radiative energy loss”. As noted, “in the current donor-acceptor system, the key factor influencing radiative/non-radiative recombination should be the HOMO offset, rather than LUMO offset.” However, in our studied systems of D18:L8-BO and D18:S-Cb, the HOMO offset of the former (0.11 eV) is actually larger than that of the latter (0.05 eV), which indicating that the larger ΔE_2 observed in the D18:S-Cb system cannot be simply attributed to a greater HOMO offset. To avoid misunderstanding, we have removed the original statement “that originated from a more obvious LUMO-LUMO energy offset between D18 and S-Cb” and prefer not to over-interpret ΔE_2 . We hope you can appreciate this clarification.

Furthermore, according to the definition of E_{loss} , ΔE_3 corresponds to the non-radiative energy loss, which is quantified as $\Delta E_3 = -kT \ln(EQE_{EL})$. This aligns with the notion that “a great solar cell also needs to be a great light-emitting diode.” The electroluminescence external quantum efficiency (EQE_{EL}) of a device is determined by the competition between the radiative recombination rate (k_r) and the non-radiative recombination rate (k_{nr}), expressed as $EQE_{EL} \propto k_r / (k_r + k_{nr})$. According to Marcus-Levich-Jortner theory, the non-radiative recombination rate is closely related to the reorganization energy (λ) of the system. A smaller reorganization energy leads to a lower non-radiative recombination rate. The reorganization energy can be expressed as $\lambda = S\hbar\omega$, where S is the Huang-Rhys factor (reflecting the electron-phonon coupling strength), \hbar is the reduced Planck constant, and ω represents the angular frequency of a

specific vibrational mode. Therefore, suppressing electron-phonon coupling can effectively reduce the reorganization energy, thereby inhibiting non-radiative recombination rate, enhancing EQE_{EL} , and thus lowering ΔE_3 . This provides a clear physical pathway for reducing non-radiative energy loss.

Both the localized excited state (LE) and the charge-transfer state (CT) describe the energy levels of excitons in which electron-hole pairs are bound, whereas the LUMO level describes the energy level of free electrons. The difference between the donor's LE state and its LUMO level equals the binding energy of the localized exciton; similarly, the difference between the CT state and the acceptor's LUMO level equals the binding energy of the CT exciton. In the schematic diagram (**Fig. R2-1**), if the binding energies of the two types of excitons are assumed equal, the LUMO offset between the donor and acceptor (ΔE_{LUMO}) would be equal to the energy difference between the donor's LE state and the interfacial CT state (ΔE_{LE-CT}). In practice, however, the binding energy of a CT exciton is usually smaller than that of an LE exciton. Consequently, ΔE_{LUMO} is typically smaller than ΔE_{LE-CT} . In our previous diagram (Fig. R4), equating the CT state to the acceptor's LUMO level and the LE state to the donor's LUMO level is a simplified treatment that neglects exciton binding energies. This simplification is intended to present the energy-level alignment more clearly for comprehension. We hope the reviewer will find this explanation acceptable.

Fig. R2-1 Schematic diagram of LUMO, LE state and CT state.

We have revised the manuscript accordingly:

It should be noted that D18:S-Cb-based OSCs obtain a slightly larger E_{loss} than that of

D18:L8-BO-based OSCs due to a larger additional radiative E_{loss} (ΔE_2).

2) In DSC measurements, the scanning temperature should typically be maintained below the onset temperature of material degradation. However, in this work, the DSC tests were conducted at temperatures approaching T_d (where the material had already undergone 5% degradation). Notably, for S-Cb, the DSC scanning temperature significantly exceeded T_d .

Reply: We sincerely appreciate your careful reading and insightful comments.

Regarding the DSC measurements, the endpoint temperature was initially set at 320 °C. Under this condition, L8-BO exhibited a complete melting peak, whereas S-Cb only showed a partial melting signal. To capture the full melting behavior of S-Cb, we increased the endpoint temperature to 340 °C and successfully observed a complete endothermic peak. However, we noted that the latter part of this melting peak exceeded the thermal decomposition temperature (T_d , 324 °C, marked by a red line in the graph). Upon closer examination, the peak shape before the red line appears symmetric, while the curve broadens noticeably after the red line. This broadening is likely attributable to additional endothermic effects from the material's thermal decomposition, which initiates once T_d is exceeded.

We are soory for not having adequately considered this factor initially, which may have led to an overestimation of the melting enthalpy. After careful re-evaluation, integrating only the melting peak region before the onset of decomposition (i.e., the region before the red line) yields a corrected melting enthalpy of 94.1 J/g. We have revised the relevant data in the manuscript accordingly.

Once again, we are grateful for your valuable comment.

Fig. 1d DSC curves of S-Cb and L8-BO (the red line is the T_d of S-Cb).

We have revised the manuscript accordingly:

However, the enthalpy change (ΔH_m) associated with the melting peak of S-Cb (94.1 J/g, determined by integrating the area of the melting peak only up to the red line, and the portion after the red line was excluded because the thermal effect in that interval includes additional heat absorption due to the thermal decomposition of S-Cb) is substantially higher than that of L8-BO (35.6 J/g).